# How Do Coding Agents Spend Your Money? Analyzing and Predicting Token Consumptions in Agentic Coding Tasks

## Abstract

AI agents offer substantial opportunities to boost human productivity across many settings. However, their use in complex workflows also drives rapid growth in LLM token consumption[1]. When agents are deployed on tasks that can require millions of tokens, two questions naturally arise: how do agents consume LLM tokens, and can we predict token usage before task execution? In this paper, we use Openhands agent as a case study and present the first empirical analysis of agent token consumption patterns using agent trajectories on SWE-bench, and further explore the possibility of predicting token costs at the beginning of task execution. We find that (1) Agent token consumption has inherent randomness even when executing the same tasks; some runs use up to $10\times$ more tokens than others. (2) Higher token usage does not lead to higher accuracy: tasks and runs that cost more tokens are usually associated with lower accuracy. (3) Unlike chat and reasoning tasks, input tokens dominate overall consumption and cost, even with token caching; and (4) while predicting total token consumption before execution is very challenging (Pearson's $r < 0.15$), predicting output-token amounts and the range of total consumption achieves weak-to-moderate correlation, offering limited but nontrivial predictive signal. Understanding and predicting agentic token consumption is a key step toward transparent and reliable agent pricing. Our study provides important empirical evidence on the inherent challenges of token consumption prediction and could inspire new studies in this direction.

## 1 Introduction

AI agents are being rapidly adopted across domains, from productivity and customer support to data analysis and software development(Liu et al., 2023c). Among these, *coding agents* are among the most widely used because they can read repositories, reason about issues, call tools, and propose patches with minimal human supervision (OpenAI, 2025; Liu et al., 2023b;c; Yang et al., 2024; Jimenez et al., 2024a). However, the prevailing pricing model for coding agents has been widely criticized for two reasons: (1) lack of transparency—users do not know the final cost until a task is finished; and (2) no guarantee of completion—users may still pay even if the task fails (Kinde, 2024). These concerns converge on a central question: **Can we predict token consumption before a task is executed?** If we could estimate token usage up front, users would better understand potential costs and choose models or strategies accordingly; providers could design clearer pricing tiers, enforce budget caps, and trigger early alerts.

In this paper, we present (to our knowledge) the first study on agent token consumption modeling and prediction. To understand the overall pattern of token usage in agentic coding workflows, we use Openhands agent Wang et al. (2025) as a case study, and conduct an empirical analysis on SWE-bench-verified (Jimenez et al., 2024b) using trajectories generated by *Claude Sonnet 3.7* and additional model backbones. We investigate three questions: (1) Do coding agents consume different amounts of tokens across tasks and runs? (2) How does task difficulty relate to token consumption? (3) Which token types (input vs. output) drive costs in sequential agent executions, and how are they distributed over the trajectory?

---

[1]We use token consumption to refer both input and output tokens used by LLM agents

Our analysis reveals three key findings. First, more complex tasks tend to consume more tokens on average, but usage varies substantially across runs; some runs use up to $10\times$ more tokens than others for the same task. Additionally, tasks and runs with more token usage are associated with lower accuracy. Second, unlike typical chat and reasoning settings, *input tokens* dominate the overall bill in agentic coding, even when token caching is enabled. A recent study, AgentTaxo (Wang et al.), reports a similar pattern in multi-agent systems, where input tokens outweigh output tokens by a factor of 2–3 times due to inter-agent communication, reinforcing that agentic workloads are broadly input-heavy across settings. Third, token consumption is not concentrated in a single step: reading long contexts (files, diffs, and retrieved artifacts) and repeated tool-mediated expansions together contribute a large fraction of the input token budget. These findings highlight both the heavy-tailed nature of token usage and the central role of context ingestion in agentic tasks.

Building on these observations, we study whether token consumption can be predicted *before* execution. We formalize a family of prediction tasks in which the goal is to estimate total, input, and output token counts from information available at the start of a run. We compare a series of input settings with the following types of information: problem statement only, anticipated tool-usage reasoning, task difficulty level; and repository information. We evaluate zero-shot and few-shot variants and consider both point predictions and range predictions (e.g., log-scale and quantile targets).

We find that providing more upfront information generally improves predictions, with the largest gains coming from tool-usage reasoning and repository information. Predicting on a log scale stabilizes training and yields better calibration of ranges. Moreover, *output-token* counts are consistently easier to predict than *input-token* counts, reflecting the high variance introduced by retrieval and context construction. While accurate *total* pre-execution prediction remains challenging (Pearson's $r < 0.15$), predicting output-token amounts and plausible consumption ranges is practical and reasonably accurate. These results suggest that providers could offer early *budget alerts* and *cost ranges* before launching expensive runs, improving transparency without overpromising point accuracy.

Overall, this work contributes:

- the first empirical characterization of token consumption in agentic coding on SWE-bench, showing large run-to-run variance and the dominance of input tokens in overall costs;
- preliminary experiments on pre-execution token-cost prediction with comprehensive model settings
- evidence that predicting output-token amounts and consumption ranges (especially in log space) is feasible but challenging, offering actionable signals for designing new pricing strategies for agentic systems.

Taken together, our empirical analysis and prediction study illuminate where tokens go in agentic coding and what can be anticipated before execution, providing concrete steps toward more predictable and user-aligned agent pricing.

## 2 DATASET

We collect agent trajectories on the SWE-bench Verified dataset (Jimenez et al., 2024b; Chowdhury et al., 2024) using the *OpenHands* framework (Wang et al., 2025), which we choose for its state-of-the-art performance among open-weight agents and its transparent, fully auditable execution pipeline. SWE-bench Verified is selected because it is the only large-scale software engineering benchmark rigorously validated by human annotators to exclude problematic or ambiguous cases.

Each problem is evaluated with four independent runs using *Claude Sonnet 3.7* (Anthropic, 2024), chosen for its strong coding performance and uniquely transparent caching and pricing policies. To assess generality, we additionally experiment with *Claude Sonnet 4* Anthropic (2025), *Qwen3-Coder-480B-A35B-Instruct* Team (2025), and *GPT-5* OpenAI (2025) with results reported in Appendix A. The collected data include full trajectories, inference logs, intermediate outputs, evaluation results, and metadata, enabling comprehensive analysis of agent behavior and cost dynamics.

In this work, we focus specifically on token consumption throughout the end-to-end problem-solving process: given an initial task description, the LLM agent autonomously interacts with the environment to finish the task without any human intervention. For each problem instance, the agent

| Metric | Description |
|---|---|
| Tool usage per tool | Average number of times each tool is used. |
| Total prompt tokens | Average number of input tokens across all rounds and runs. |
| Total completion tokens | Average number of output tokens across all rounds and runs. |
| Cache creation input tokens | Average number of input tokens written into cache. |
| Cache read input tokens | Average number of tokens retrieved from cache. |
| Cost per round / total | Monetary cost values returned directly by the API. |

Table 1: Extracted metrics from LLM completion logs.

proceeds in multiple rounds: in each round, the LLM generates a response based on the current prompt, followed by a tool call and execution. In particular, the full conversation history, including all previous prompts and completions, is carried forward unchanged into subsequent rounds.

To enable a more detailed analysis of LLM behavior and its corresponding token consumption during problem-solving, we extract a set of fine-grained metrics from the LLM completion history. These metrics are obtained by parsing the structured JSON outputs of the agent and leveraging the usage information, which records all LLM interactions at each round. Together, the extracted metrics capture both the functional behavior of the agent, such as tool usage and file access patterns, as well as the underlying token-level dynamics. For all token-related metrics, we report values averaged over the four independent runs per problem. Table 1 summarizes the extracted features and their definitions.

# 3 An Empirical Investigation of Agent Token Consumption

## 3.1 Variability in Token Consumption and Tool Usage

**Variations across different problems and different runs** We begin by analyzing the variation in token consumption and tool usage across different problems (averaged over four independent runs) and across different runs for the same problem. For both total prompt tokens and total completion tokens, we observe substantial variation across problems, indicating a high degree of instability in token usage depending on the specific problem instance.

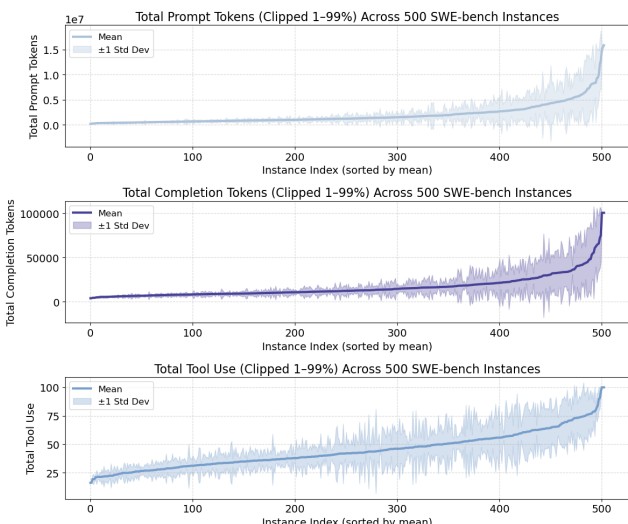

Figure 1: **Distribution of token and tool usage across SWE-bench instances.** Each curve shows the mean and standard deviation of total prompt tokens, completion tokens, and tool usage.

To further investigate this phenomenon, we sort all problems by their average total token cost (from low to high) and visualize the variation in both token usage and tool usage across the four runs for each problem (Figure 1). We observe that problems with higher overall token costs tend to exhibit substantially larger cross-run variance, indicating that the agent's behavior becomes increasingly unstable on more complex or longer tasks that require more tokens to solve. A similar pattern holds

for tool usage: although the trend is less pronounced than for token consumption, notable variability still exists both across different problems and across runs of the same problem.

**Accuracy and the Inverse Test-time Scaling Paradox**  We analyze how accuracy varies with token and tool usage across problems and cost levels. Figure 2(a) shows that accuracy tends to decrease as the average token or tool consumption increases. Problems that demand more resources often correspond to cases where the agent explores multiple unproductive trajectories, leading to longer contexts and reduced precision in reasoning.

To further examine this trend, Figure 2(b) stratifies accuracy by within-problem cost levels. Each problem's four runs are ranked by total cost and grouped into *MinCost*, *LowerCost*, *UpperCost*, and *MaxCost*. Interestingly, accuracy rises slightly from *MinCost* to *LowerCost* but then declines for higher-cost bins, showing the clearest drop at *MaxCost*. This pattern represents an *inverse test-time scaling* phenomenon Snell et al. (2024); Wu et al. (2025), where greater resource usage does not yield better outcomes. Instead, excessive compute often reflects inefficient reasoning cycles and context bloat that hinder rather than enhance performance.

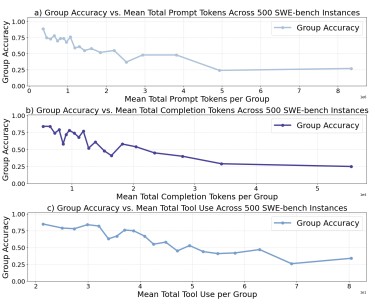

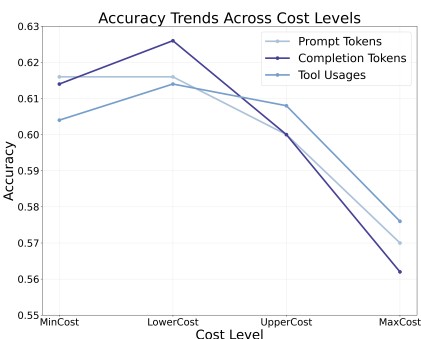

(a) Group accuracy vs. mean token and tool usage across 500 SWE-bench instances.

(b) Accuracy trends across cost levels.

Figure 2: **Accuracy variation across usage and cost levels.** (a) Accuracy as a function of average resource usage per group. (b) Accuracy stratified by within-problem cost levels.

Motivated by these observations, we further examine the behavioral patterns underlying high-cost failures by analyzing *distinct* and *repeated* file *view* and *modify* actions across cost levels. As shown in Figure 3, both viewing and editing activity increase with cost, but the growth is driven primarily by a sharp rise in *repeated* actions rather than distinct ones. This suggests that many expensive but failed runs are characterized by redundant back-and-forth file access and re-editing, reflecting inefficient search dynamics that inflate context length and token usage without proportional progress. While not all high-cost runs are dominated by redundancy, this pattern provides a concrete behavioral explanation for the inverse accuracy–cost relationship observed above.

Notably, in our GPT-5 experiments, the inverse test-time scaling effect is noticeably weaker (Figure 20 from *MinCost* to *UpperCost*), with a smaller increase in repeated file operations as cost rises. This indicates that when expensive runs involve less back-and-forth repetition, the associated performance degradation under higher compute is correspondingly milder.performance degradation under higher compute.

**Human vs. Model Perception of Difficulty**  We analyze how problem difficulty affects token consumption and tool usage. The difficulty levels follow the SWE-bench Verified dataset Jimenez et al. (2024b); Chowdhury et al. (2024), which categorizes problems based on the estimated time required by professional developers to resolve them (e.g., "<15 min", "15 min – 1 hour", "1–4 hours", ">4 hours"). Because there are only three instances in the >4 hours category, we merge it with the 1–4 hours group and report them together as >1 hours.

Figure 4 shows the distribution of prompt tokens, completion tokens (scaled by 100 for visibility), and tool usages across difficulty levels. While overall resource consumption tends to rise with problem difficulty, the relationship is far from linear. Notably, **10.31%** of tasks labeled as "<15-minute" required more total tokens than the average ">1-hour" instance, and **24.44%** of ">1-hour" tasks consumed fewer tokens than the "<15-minute" group.

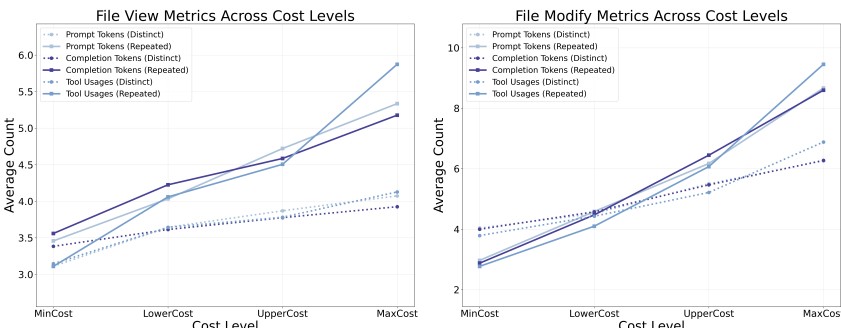

Figure 3: **Distinct vs. repeated file operations across cost levels.** Higher-cost runs show a disproportionate increase in repeated accesses and edits, indicating redundant back-and-forth behavior that drives inefficiency and excessive token usage.

These outliers highlight that human-estimated difficulty does not always align with the model's internal notion of complexity. Tasks that seem easy to humans may still demand extensive reasoning, exploration, or tool interaction from the model, whereas some "hard" problems may be efficiently solvable given the model's prior knowledge or search strategies. Consequently, human-labeled difficulty is an imperfect predictor of model resource expenditure.

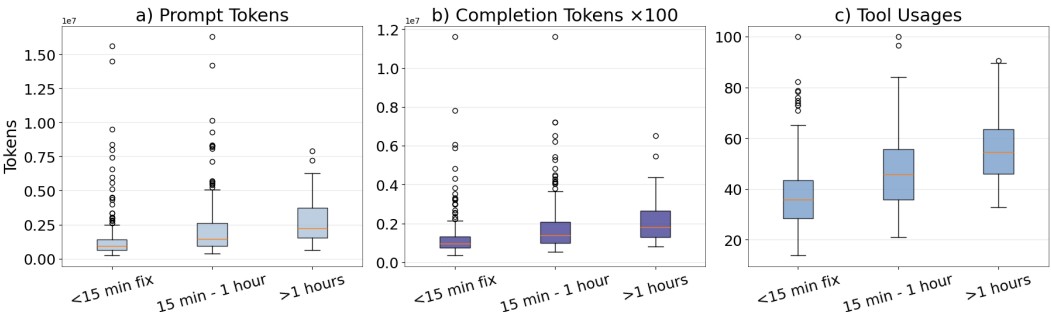

Figure 4: **Token and tool usage across difficulty levels.** Harder tasks generally consume more resources, though some easy ones still show high token usage, revealing large behavioral variance.

### 3.2 TOKEN–COST DYNAMICS ACROSS PHASES AND ROUNDS

To understand how different token types contribute to overall cost, we analyze the relationship between token usage and cost both at an aggregate phase level and within a representative round-level case study.

**Charging scheme** The total cost for each round of interaction returned from Claude API is composed by separately accounting for four categories of tokens: (i) non-cached prompt tokens, (ii) completion tokens, (iii) cache creation tokens, and (iv) cache read tokens. Non-cached prompt tokens correspond to input text that is processed without leveraging the cache, whereas cache tokens are either created (written once to enable future reuse) or read (retrieved in later rounds at a lower marginal cost). Each category is billed at a distinct rate, with cache creation depending on the persistence setting (here we use the 5-minute write rate). The total per-round cost is simply the sum of these components.[2]

**Phase-level Token Usage Dynamics** We divide each problem-solving trajectory into five chronological phases—*Early*, *Early-Mid*, *Mid*, *Later-Mid*, and *Later*—based on the total number of interaction rounds for each problem. Within each phase, we aggregate data across 500 problem instances and compute per-round statistics, including the correlation between token types and cost, as well as the average proportion of each token type in both token count and cost.

---

[2]Pricing details are available at `https://docs.claude.com/en/docs/about-claude/pricing`. The cost calculation is shown in Appendix B.

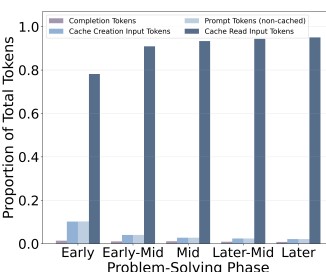 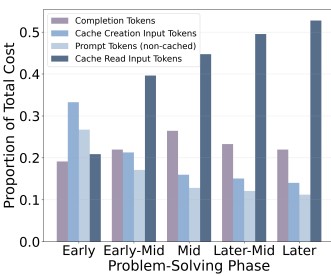 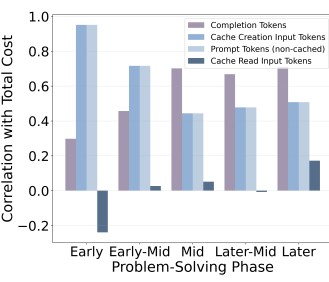

(a) Proportion of total tokens.  (b) Proportion of total cost.  (c) Correlation with total cost.

Figure 5: **Token usage and cost composition across problem-solving phases.** Each phase (*Early* → *Later*) represents an equal partition of the agent's trajectory. (a) Token composition remains stable across phases, with non-cached prompt and cache-read tokens dominating total token usage. (b) Cost composition shows that cache creation tokens dominate in the early stages, while cache-read costs become the primary contributor in later phases. (c) Correlation analysis shows that non-cached prompt and cache creation tokens dominate early phases, while completion tokens are most correlated with total cost later.

Building on this setup, we analyze how different token types contribute to cost and usage throughout the agent's problem-solving trajectory. The overall token composition remains stable across phases (Figure 5a), with cache-read tokens accounting for most token usage. This stability suggests that although the total number of tokens fluctuates, the fundamental interaction structure remains consistent throughout the process. Notably, the token count proportions of non-cached prompt tokens and cache creation tokens remain nearly identical across all five phases, reflecting that whenever new (non-cached) prompt tokens appear, they are subsequently cached for reuse in subsequent rounds.

The proportion of cache-read tokens in total cost increases steadily across phases (Figure 5b), yet their correlation with total cost remains consistently low (Figure 5c). This indicates that cache reads are economical operations: as their cost share rises, it reflects efficient reuse of cached information rather than increased expenditure. In other words, greater reliance on cache reads corresponds to cheaper progress rather than inflated computation.

Finally, as shown in Figure 5, cache creation and non-cached prompt tokens dominate the early stages, when the agent constructs its working context. Completion tokens become increasingly influential from the mid to later phases, showing the strongest correlation with total cost and reflecting the token-intensive nature of reasoning and code generation.

**Round-level cost breakdown.** To complement the phase-level aggregate view, we now zoom into a representative case study. Figure 6 presents the exact per-round cost decomposition across the four token categories. We observe that the cost of *non-cached prompt tokens* closely mirrors that of *cache creation tokens*, which is expected given their similar per-token rates. Moreover, whenever a segment of non-cached prompt tokens appears, it is typically followed by corresponding cache creation, reflecting the mechanism by which uncached input is soon persisted for reuse. The *cache read tokens* accumulate steadily over rounds and constitute a large fraction of the total cost; however, the total round cost does not monotonically increase, explaining the low correlation between cache reads and per-round cost observed in Figure 5c. Consistent with the trends highlighted earlier, the dominant cost drivers vary across phases: in the *Early* phase, cache creation dominates total cost, while in the *Later* phases, completion tokens become the primary contributor as generation expands.

**Trajectory behavior analysis** We selected six representative rounds and inspected their detailed trajectories to understand how the agent balances tool use, code generation, and verification. A compact summary of the representative steps is shown in Table 2.

*Rounds dominated by non-cached prompt tokens.* In three rounds, token usage was dominated by uncached input from tool calls. The agent relied heavily on repository exploration and targeted searches (e.g., using `grep` or signature inspection) to locate relevant code regions and align them with stack traces. These actions generated large uncached input dumps, while the completions were relatively short, focusing on synthesizing evidence rather than producing long code edits.

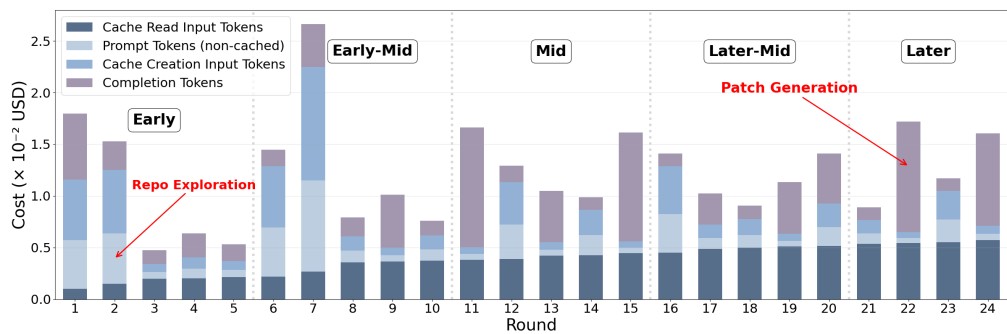

Figure 6: **Round-level token cost breakdown.** Exact per-round costs for Claude Sonnet 3.7, decomposed into non-cached prompt tokens, cache creation (5m persistence), cache reads, and completion tokens. The dominant cost source shifts from cache creation in early phases to completion tokens in later phases.

*Rounds dominated by completion tokens.* In the other three rounds, completion tokens dominated due to long generated outputs. The agent created external reproducers, drafted minimal harnesses, implemented guarded fixes (e.g., for `None` return annotations), and validated results with environment overrides. These trajectories involved extensive explanatory text and multi-line code edits, with most tokens coming from generated content rather than tool outputs.

| Round | Tool Usage | Action summary |
|---|---|---|
| 1 | execute_bash (ls/grep) | Locates decorator, call sites, and links trace. |
| 2 | execute_bash (repo search) | Explores files broadly, traces errors. |
| 7 | execute_bash (signature search) | Dumps large files while checking signatures. |
| 11 | str_replace_editor | Writes reproducer, explains failure, suggests fix. |
| 15 | str_replace_editor | Proposes patch for `None` return issue. |
| 22 | str_replace_editor | Adds guard for `None` and verifies fix. |

Table 2: Representative trajectory steps across analyzed rounds.

## 4 PREDICTING TOKEN CONSUMPTION

In this section, we investigate approaches for predicting the token consumption of the OpenHands agent. Prediction quality is measured using the Pearson correlation coefficient.

### 4.1 SINGLE LLM PREDICTION

Given the inherent variability and stochasticity of agent trajectories, predicting token consumption is challenging. At the same time, the analyses in the previous section suggest that token usage patterns encode rich information that can be leveraged. Motivated by this, we directly prompt a large language model to estimate token cost. Specifically, we employ Claude Sonnet, the same model that powers the OpenHands agent, to perform this prediction task.

**Prediction Settings** We experimented with four basic prompting configurations, each differing in the richness of the information provided, and additionally compare prediction in exact numbers versus log scale, as well as in zero-shot and few-shot settings. The four settings are summarized in Table 3. We run each setting five times and report the average correlation together with the variance across the five runs.

| Setting | Description |
|---|---|
| P | Problem statement only |
| PT | Problem statement + tool usage reasoning |
| PTD | Problem statement + tool usage reasoning + difficulty level |
| PTDR | Problem statement + tool usage reasoning + difficulty level + repository information |

Table 3: Prompt configurations for single LLM token prediction.

The *PTDR* configuration requires special handling due to the length of GitHub repository structures and the limited context window of Claude. We therefore adopt a two-step prompting strategy. In the first step, we extract the repository file tree and file-level statistics using the GitHub API, and feed both the problem statement and this raw repository metadata to the LLM. The model then produces a concise summarization of repository information relevant to the problem statement. In the second step, we integrate this summarization with the other inputs (problem statement, tool usage reasoning, and difficulty level) and perform the final prediction.

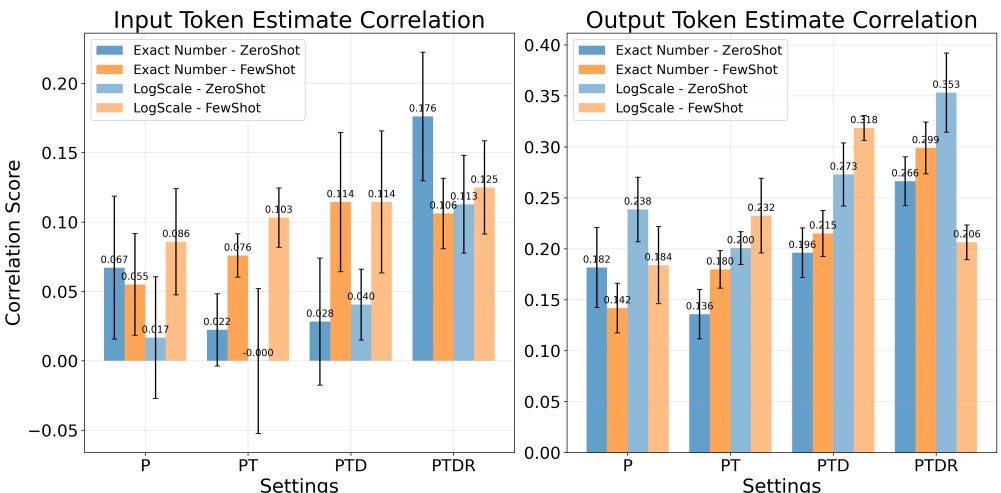

Figure 7: **Prediction performance across settings for single LLM prediction.** Average correlation (with variance) for input and output token prediction under different settings. Log-scale prediction consistently outperforms exact-number prediction, and repository information proves especially valuable.

**Results and Observations**    As shown in Figure 26, the prediction of prompt tokens exhibits substantially higher variance than the prediction of output tokens, indicating that input token estimation is considerably more difficult. This is intuitive given the implementation of the coding agent: all past conversation history is concatenated into the prompt of the next round, leading to a cumulative and approximately linear increase in prompt length. Such accumulation makes the prompt token sequence less stable and more challenging to predict accurately.

By contrast, the prediction of output tokens is more stable, with smaller variance and consistently higher correlation across settings. Overall, richer context improves prediction performance: models with more information achieve higher correlation. Predictions based on log-scale transformation are also generally more accurate than predictions of exact numbers, suggesting that relative magnitudes are easier to capture than absolute counts. Interestingly, zero-shot predictions with repository information (RI) outperform few-shot predictions without repository information, for both log-scale and exact number settings, underscoring the strong value of structural repository knowledge.

Nevertheless, the overall task of token prediction remains challenging due to the interplay of multiple factors, including cumulative context, stochastic agent behavior, and the bias induced by few-shot example selection. We discuss these challenges in greater depth in Section 4.3.

## 4.2 SELF-PREDICTION BY THE SAME AGENT

Instead of relying solely on a single LLM call for prediction, we also experimented with using the coding agent itself as the predictor. This approach is motivated by the natural intuition that the coding agent, being the one that would actually execute the task, has the most detailed knowledge of its own workflow. In this setting, we modify the instructions of the OpenHands coding agent so that it predicts token consumption rather than solving the coding problem. The agent retains its full tool-calling and interaction capabilities, but its task is modified so that instead of implementing a solution, it is guided by an updated system prompt to estimate the token cost required for the given

task. This setup allows the agent to actively explore the problem's codebase, generate intermediate plans, and reason about the likely workload.

**Prediction Settings**    We experimented with two settings. In the basic setting, the agent receives the instruction and no further guidance (see Appendix E.1 for the full prompt). In the fine-grained setting, the agent is explicitly instructed to make more fine-grained predictions and to decompose the task (see Appendix E.2 for the full prompt). Due to budget limitations, we conducted three independent runs for each setting. Each run used the same 500 samples with the previous setting. Because prediction itself consumes tokens, the cost of running the predictor is a non-negligible factor; we therefore report the token overhead of one run in Appendix F.

**Results and Observations**    From Figure 8, we observe that using the coding agent itself as the predictor achieves reasonable performance in both settings, with non-trivial correlation scores for both input and output token estimation. When we add the fine-grained instruction, both input and output token prediction improve noticeably compared to the basic setting. We also find that the variance of output-token prediction is considerably larger than that of input-token prediction, likely because the number of output tokens is typically much smaller and thus more variable. Introducing the fine-grained instruction significantly reduces this variance, making the output-token estimates more stable.

**Self-Prediction Cost vs. Task Actual Cost and Prediction Error**    We further examine whether the cost incurred by self-prediction itself is informative of the true task cost and whether it affects prediction quality. As detailed in Appendix C, we find a weak linear correlation but a moderate rank-based correlation between self-prediction cost and actual task cost, indicating that self-prediction better captures *relative* cost ordering than precise magnitudes. Importantly, we observe no meaningful correlation between self-prediction cost and prediction error across prompt, completion, or total tokens. This suggests that higher self-prediction expense does not translate into improved or degraded prediction accuracy. Together, these results indicate that self-prediction provides a low-cost, coarse-grained signal of task difficulty and resource demand without introducing systematic bias into the token estimates.

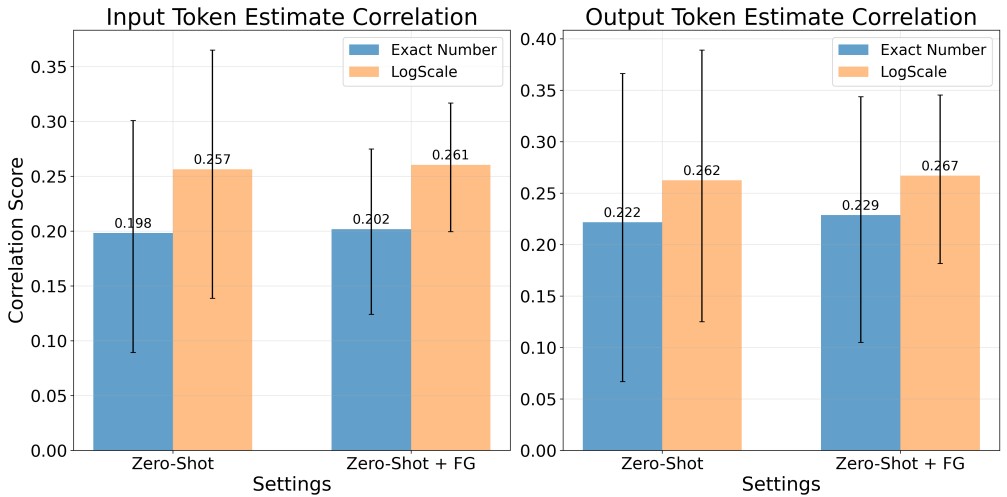

Figure 8: **Prediction performance across settings for self-prediction by the same agent.** Similar to single LLM setting, We find that log-scale prediction performs better. Besides, output tokens benefit more from fine-grained setting.

## 4.3    CHALLENGES

**Single LLM Prediction**    Despite the encouraging results, two factors complicate reliable prediction. First, our **few-shot setup**, selecting four examples per difficulty level, can introduce bias, since each run exhibits high variance and small sampling choices may skew outcomes. Second, the configuration we initially expected to perform best, which combines all sources of information, in fact

showed weaker results. This suggests that **excessive detail may overwhelm the model**, especially when the prediction target is only a coarse log-scale number rather than a fine-grained output.

**Self-Prediction by the Same Agent**  While it is natural to let a coding agent predict its own token consumption—since it is arguably most familiar with its own decision-making—we find that this approach, despite yielding higher correlation with actual usage, suffers from several challenges. In both settings we have explored, self-prediction tends to consistently overestimate the true cost. We also considered providing few-shot examples to give the agent a better sense of the token cost per round. However, it is difficult to construct representative high-quality examples: available samples are heterogeneous and often fail to capture general patterns, and directly including multi-round few-shot traces would quickly exceed the context window of current LLMs. Therefore, few-shot settings require further study. Moreover, compared to the cost of a single LLM call, users may be unwilling to tolerate the latency and overhead introduced by multi-round estimation. As a result, self-prediction, although appealing in principle, remains challenging in practice.

## 5 DISCUSSION

In this paper, we present the first empirical analysis of agent token consumption and explore different methods to predict it. In this section, we highlight the main limitations of our study and discuss the implications of our findings for the design and pricing of agent-based systems.

**Limitations**  A key limitation of our study lies in the range of model backbones evaluated. Although we analyze multiple models, including Claude Sonnet 3.7, Sonnet 4, and Qwen3 Coder, this still covers only a narrow slice of the broader agentic model landscape. Collecting full trajectories for additional models is extremely costly in both time and money, which constrains the breadth of our evaluation. While the patterns we observe are consistent across the models tested, further validation on more architectures would strengthen the generality of our findings. To support such extensions, we release our experimental pipeline so future work can replicate and expand our analysis.

**Agent Pricing**  One of the central challenges for providers of agentic systems is how to price them. Traditional AI products such as ChatGPT often rely on subscription-based models, since typical users consume only a limited number of tokens. By contrast, agentic tasks can require very large amounts of tokens, even for seemingly simple problems. This makes prediction of token consumption essential for designing sustainable pricing strategies. Our findings show that token usage, especially input tokens, is highly variable and difficult to predict due to the randomness of agent trajectories. As a result, consumption-based pricing may remain the most practical option until we figure out better ways to predict token usage.

**User Transparency**  Reliable token predictions are also important for user transparency. Ideally, an agentic system would be able to inform users about the likely cost of a task before execution. However, current language models struggle to provide accurate estimates, limiting the ability to present exact costs in advance. That said, our results suggest that predictions on a log scale perform better, which could help systems flag tasks with potentially high costs. This approach may allow providers to alert users about potentially large charges and ask for user permissions before proceeding, even if precise predictions are not possible.

## 6 CONCLUSION

With the rapid growth of agent token consumption in various settings, predicting future token usage before task execution becomes an important task for reasonable and transparent pricing of AI agents. In this paper, we propose the first study aimed at understanding and predicting agent token consumption on agentic coding tasks. Through a series of empirical analyses of agent trajectories, we reveal important findings about the pattern of agent token usage in agentic coding tasks. Furthermore, we explore a range of token usage prediction methods and highlight the challenges of predicting agent token consumption before task execution.

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

## A GENERALIZATION ACROSS ALTERNATIVE MODEL BACKBONES

To assess the generalizability of our observations, we evaluate the coding agent's behavior using alternative model backbones[3]. Despite differences in pricing, caching mechanisms, and architecture, the core phenomena: **large run-to-run variability, heavy-tailed token consumption, and the inverse test-time scaling paradox**, consistently appear across models.

We also observe substantial variance in resource usage across human-defined difficulty levels, further showing that human-perceived difficulty is a poor predictor of an agent's computational effort: many "easy" tasks incur high token costs, while some "hard" tasks consume fewer resources.

Across problem-solving phases all models exhibit a similar progression: tokens most strongly correlated with cost shift from input-related tokens during early repository exploration to completion tokens during later patch-generation. At the same time, prompt-related and cache-read tokens remain the dominant contributors to both total token volume and cost. These consistent cross-model patterns reinforce the robustness of our overall findings.

### A.1 SONNET 4

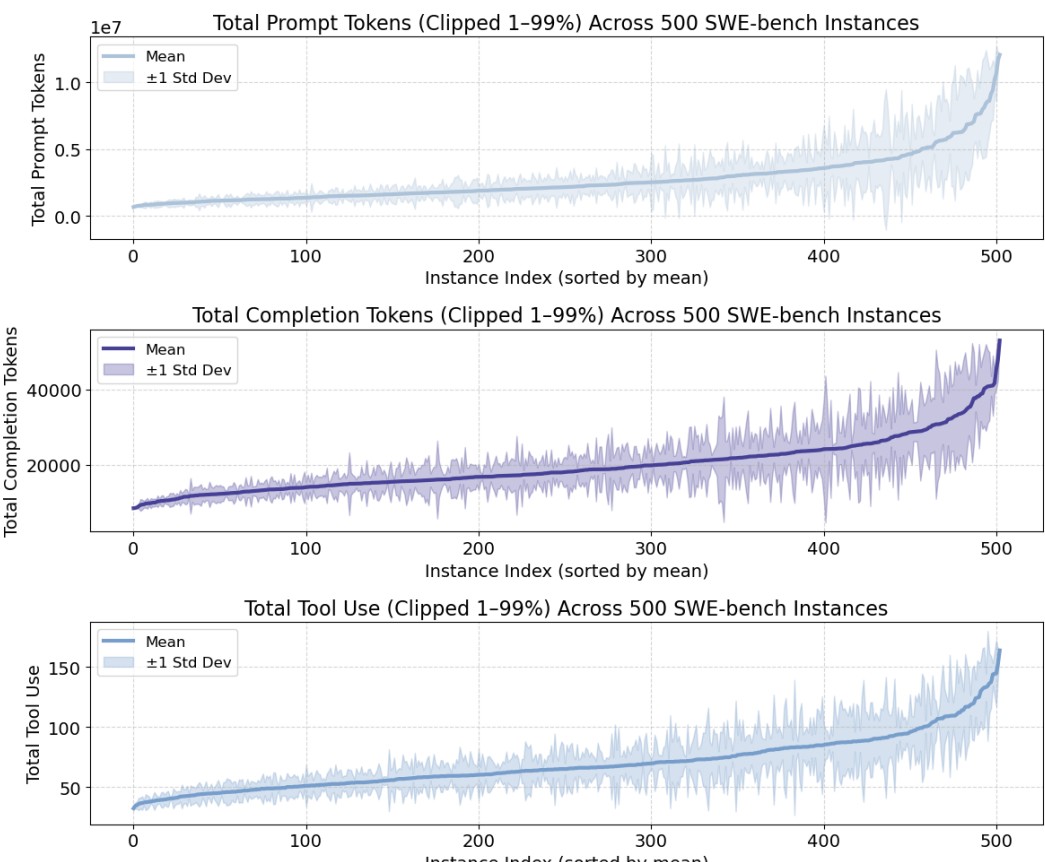

Figure 9: Distribution of token and tool usage across SWE-bench instances.

---

[3]Qwen3-Coder-480B-A35B-Instruct is an open-source model. For consistency and ease of comparison, we adopt the pricing scheme of Qwen3-Coder-Plus when conducting our cost analysis.

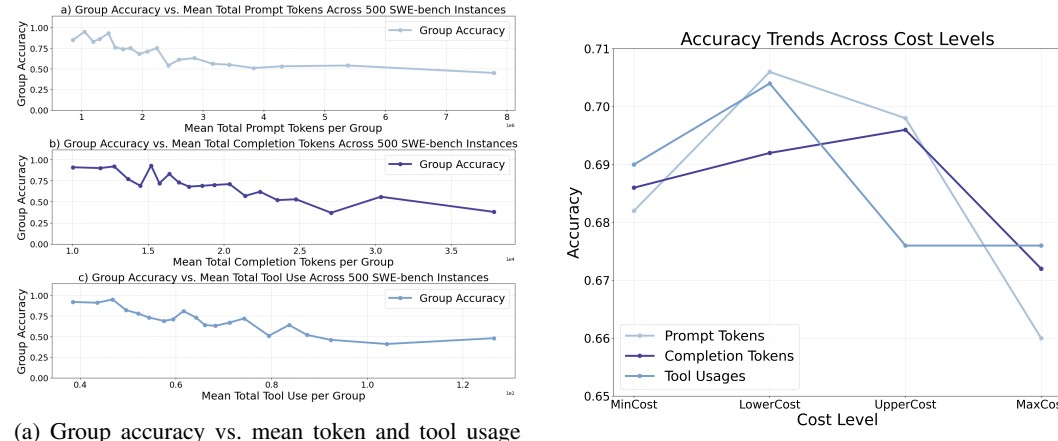

(a) Group accuracy vs. mean token and tool usage across 500 SWE-bench instances.

(b) Accuracy trends across cost levels.

Figure 10: **Accuracy variation across usage and cost levels for Sonnet 4.** (a) Accuracy as a function of average resource usage per group. (b) Accuracy stratified by within-problem cost levels.

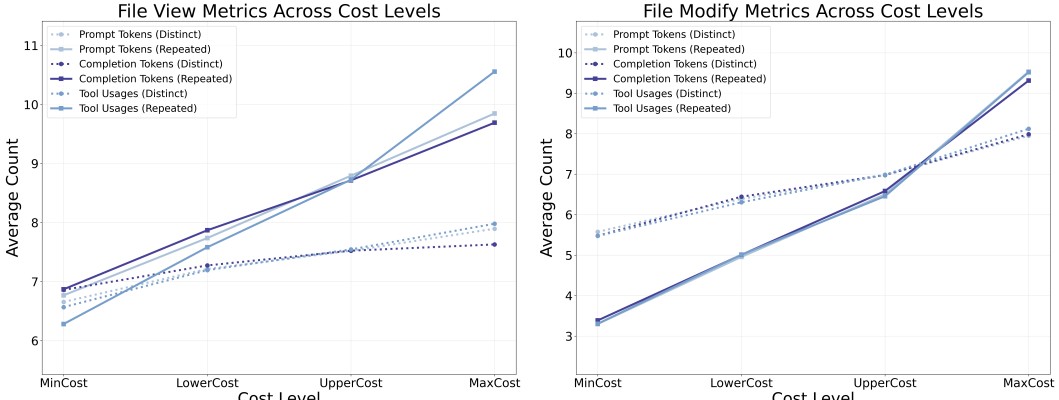

Figure 11: Distinct vs. repeated file operations across cost levels for Sonnet 4.

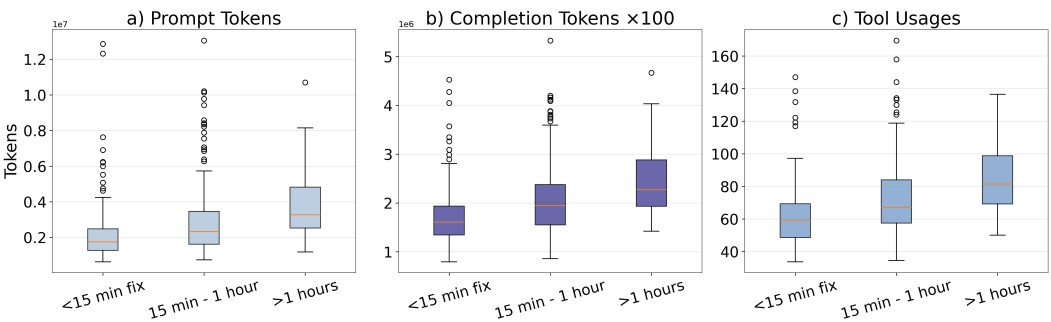

Figure 12: Token and tool usage across difficulty levels for Sonnet 4.

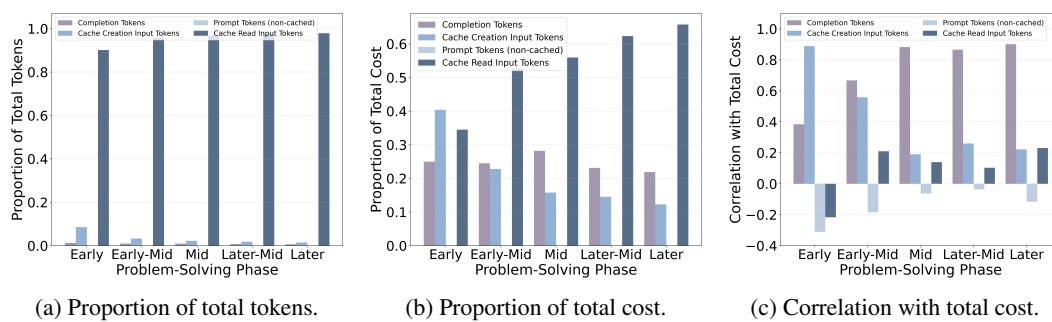

(a) Proportion of total tokens.  (b) Proportion of total cost.  (c) Correlation with total cost.

Figure 13: Token usage and cost composition across problem-solving phases for Sonnet 4.

## A.2  QWEN3-CODER-480B-A35B-INSTRUCT

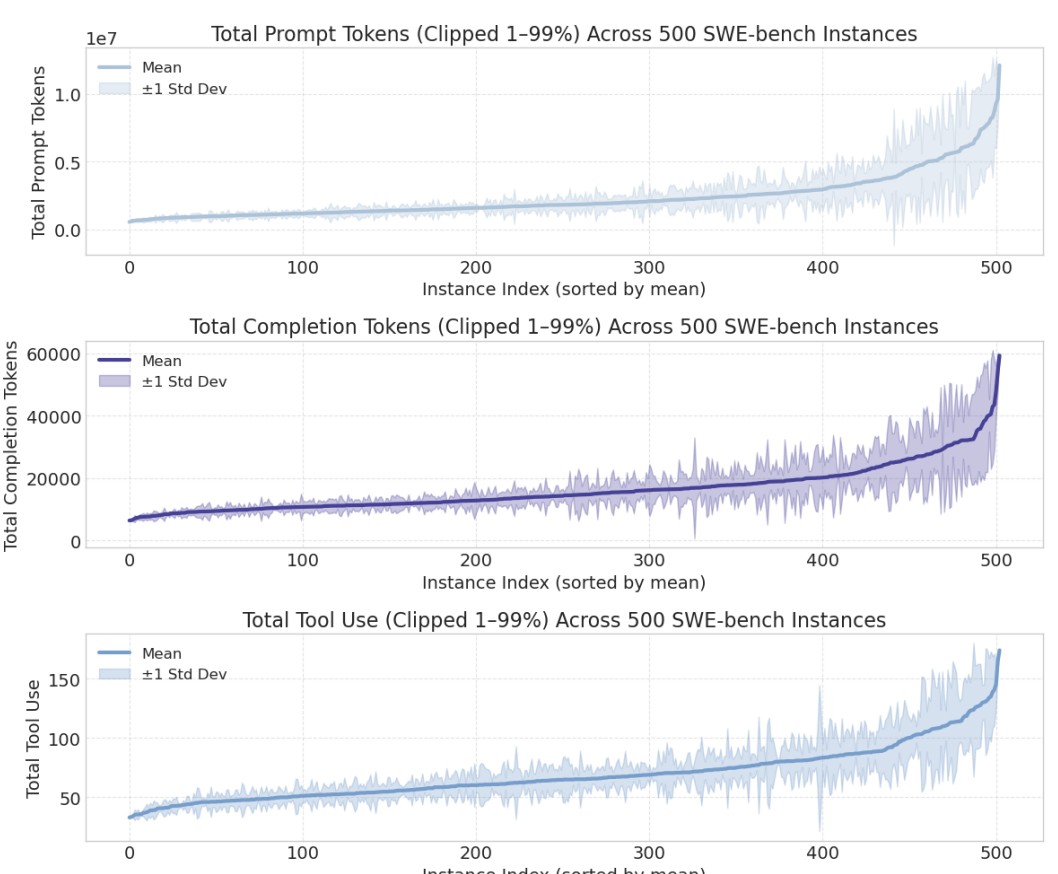

Figure 14: Distribution of token and tool usage across SWE-bench instances.

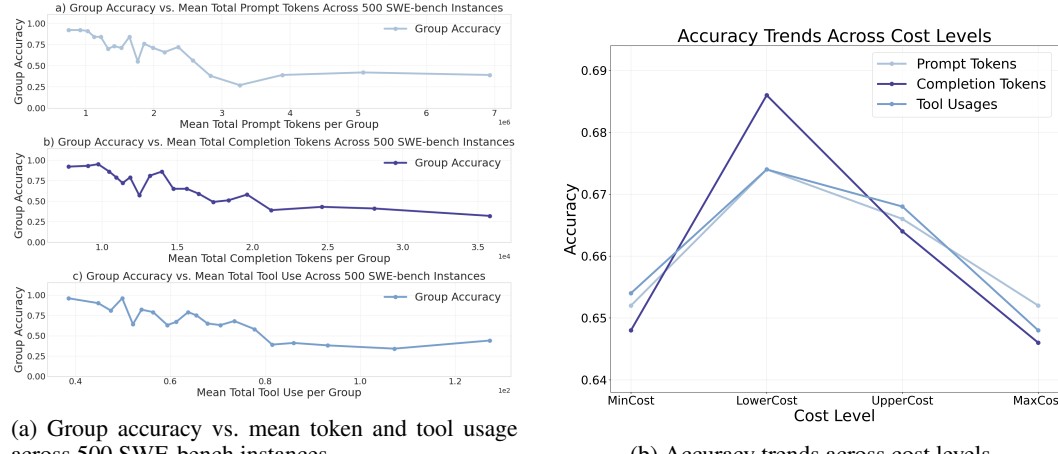

(a) Group accuracy vs. mean token and tool usage across 500 SWE-bench instances.

(b) Accuracy trends across cost levels.

Figure 15: **Accuracy variation across usage and cost levels for Qwen3-Coder-480B-A35B-Instruct.** (a) Accuracy as a function of average resource usage per group. (b) Accuracy stratified by within-problem cost levels.

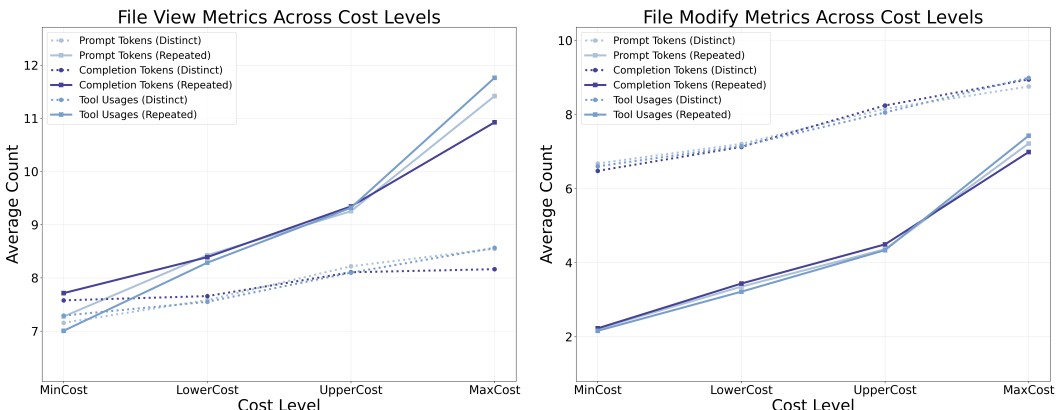

Figure 16: Distinct vs. repeated file operations across cost levels for Qwen3-Coder-480B-Instruct.

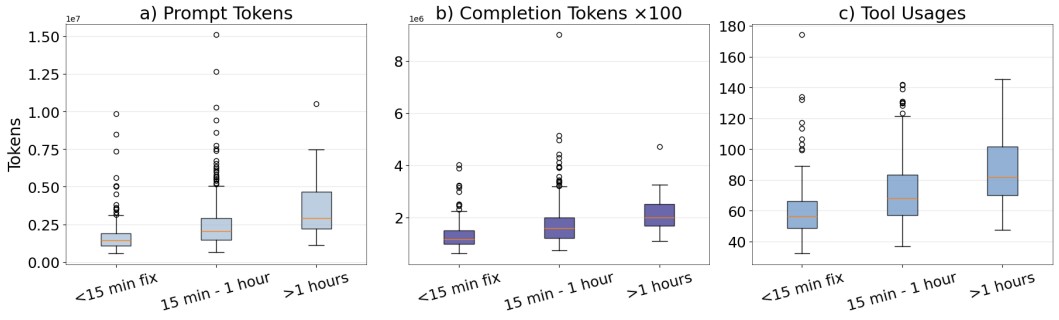

Figure 17: Token and tool usage across difficulty levels for Qwen3-Coder-480B-A35B-Instruct.

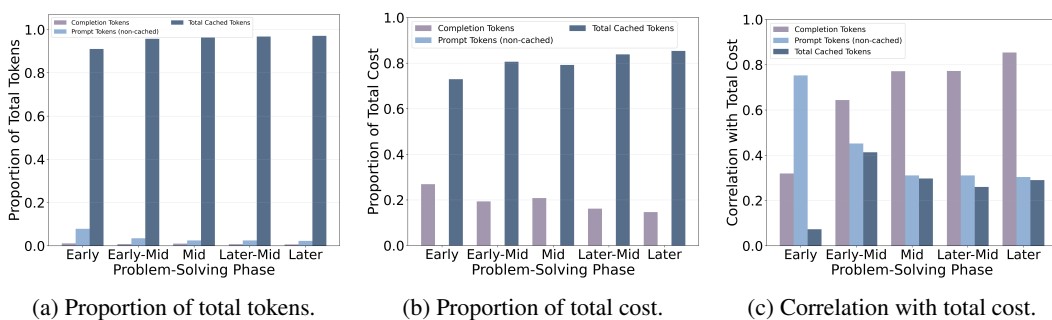

(a) Proportion of total tokens.     (b) Proportion of total cost.     (c) Correlation with total cost.

Figure 18: Token usage and cost composition across problem-solving phases for Qwen3-Coder-480B-A35B-Instruct.

## A.3 GPT-5

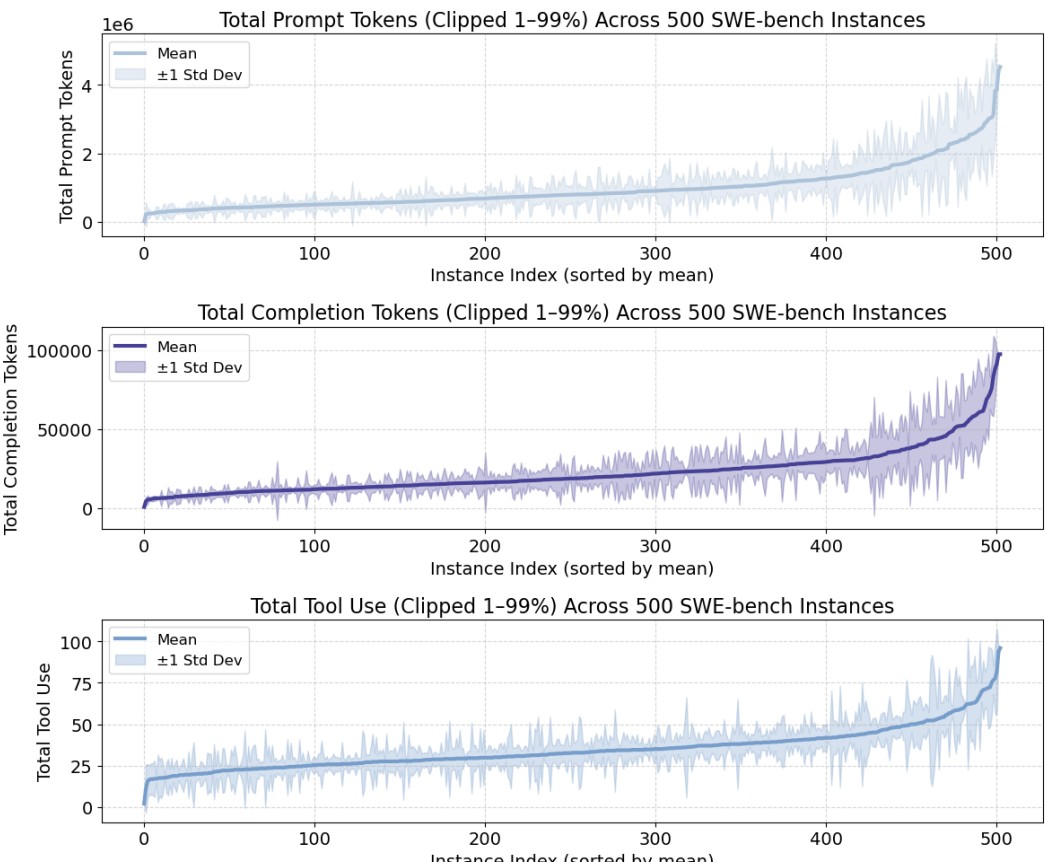

Figure 19: Distribution of token and tool usage across SWE-bench instances.

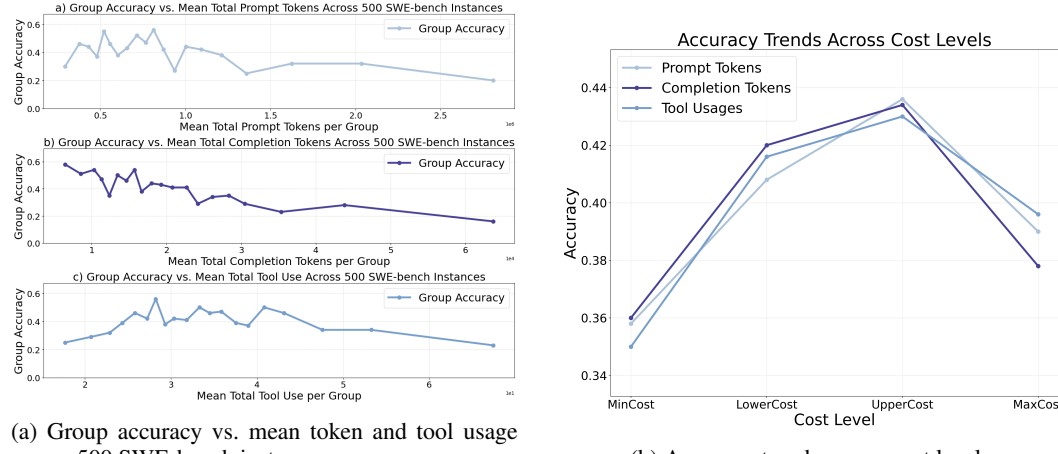

(a) Group accuracy vs. mean token and tool usage across 500 SWE-bench instances.

(b) Accuracy trends across cost levels.

Figure 20: **Accuracy variation across usage and cost levels for GPT-5.** (a) Accuracy as a function of average resource usage per group. (b) Accuracy stratified by within-problem cost levels.

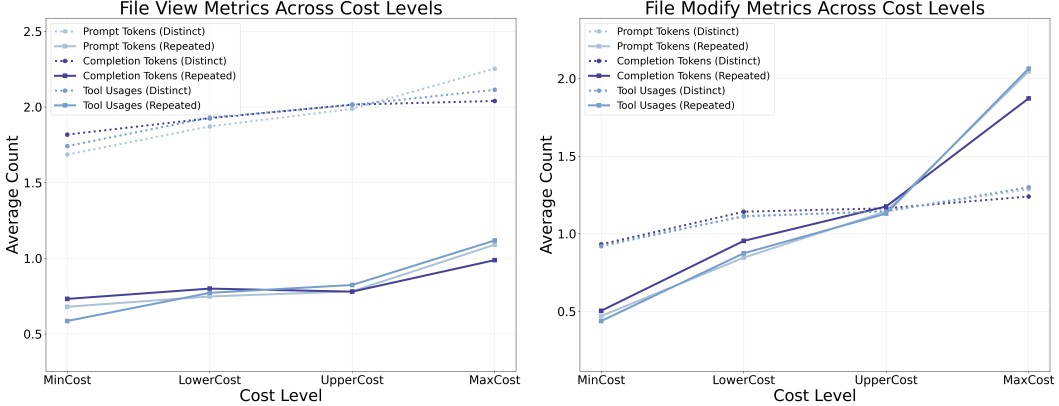

Figure 21: Distinct vs. repeated file operations across cost levels for GPT-5.

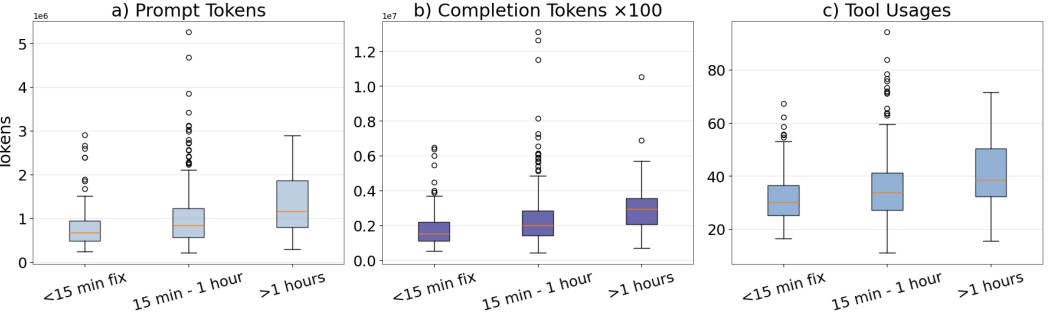

Figure 22: Token and tool usage across difficulty levels for GPT-5.

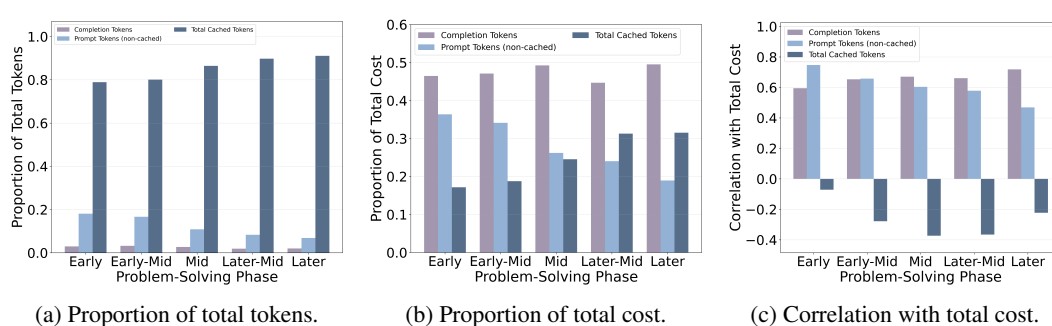

(a) Proportion of total tokens.  (b) Proportion of total cost.  (c) Correlation with total cost.

Figure 23: Token usage and cost composition across problem-solving phases for GPT-5.

# B  COST CALCULATION DETAILS

## B.1  CLAUDE MODELS

$$\text{Prompt}_{\text{non-cached}} = \text{Prompt}_{\text{total}} - \text{CacheRead}. \tag{1}$$

$$
\begin{aligned}
\text{Cost}_{\text{round}} = &(\text{Prompt}_{\text{non-cached}} \times r_{\text{in}}) \\
&+ (\text{Completion} \times r_{\text{out}}) \\
&+ (\text{CacheCreation} \times r_{\text{cache\_create}}) \\
&+ (\text{CacheRead} \times r_{\text{cache\_read}}).
\end{aligned}
\tag{2}
$$

where $r_{\text{in}}$ is the base input rate, $r_{\text{out}}$ is the output rate, $r_{\text{cache\_create}}$ is the cache creation rate (5-minute writes in our setting), and $r_{\text{cache\_read}}$ is the cache read rate.

## B.2  QWEN MODELS (IMPLICIT CACHE ONLY)

For Qwen models (qwen3-coder-plus and qwen3-coder-flash series), we use only the implicit cache mechanism. The API reports the number of prompt tokens served from the implicit cache, and we ignore explicit cache creation since it is not used in our setting.

Qwen applies tiered pricing based on the total prompt tokens in each request.[4] For qwen3-coder-plus, the input and output costs (per million tokens) are:

| Token Range | Input Cost | Output Cost |
|---|---|---|
| $0 < \text{tokens} \leq 32\text{K}$ | $1 | $5 |
| $32\text{K} < \text{tokens} \leq 128\text{K}$ | $1.8 | $9 |
| $128\text{K} < \text{tokens} \leq 256\text{K}$ | $3 | $15 |
| $256\text{K} < \text{tokens} \leq 1\text{M}$ | $6 | $60 |

Let $\text{Prompt}_{\text{total}}$ be the total prompt tokens, $\text{CacheRead}_{\text{implicit}}$ the portion served from the implicit cache, and Completion the number of output tokens. Let $r_{\text{in}}$ and $r_{\text{out}}$ denote the tier-specific input and output rates. Implicit cache reads are billed at $r_{\text{cache\_read}}^{\text{implicit}} = 0.2\, r_{\text{in}}$.

The non-cached prompt is

$$\text{Prompt}_{\text{non-cached}} = \text{Prompt}_{\text{total}} - \text{CacheRead}_{\text{implicit}}. \tag{3}$$

The total cost is

$$
\begin{aligned}
\text{Cost}_{\text{round}} = &(\text{Prompt}_{\text{non-cached}} \times r_{\text{in}}) \\
&+ (\text{CacheRead}_{\text{implicit}} \times 0.2\, r_{\text{in}}) \\
&+ (\text{Completion} \times r_{\text{out}}).
\end{aligned}
\tag{4}
$$

## B.3  GPT-5 MODELS (IMPLICIT CACHE)

For GPT-5 models, we use OpenAI's implicit caching mechanism. The API reports cached prompt tokens automatically, without explicit cache creation. At the time of our experiments, the official pricing is: *Input: $1.250 / 1M tokens*, *Cached input: $0.125 / 1M tokens*, and *Output: $10.000 / 1M tokens*.[5]

The cost calculation follows the same formulation as for the Qwen models (Section B.2).

---

[4]See the official Model Studio pricing documentation:https://www.alibabacloud.com/help/en/model-studio/models#8e453767fbkka

[5]See the official pricing documentation: https://openai.com/index/introducing-gpt-5/

## C ANALYSIS OF SELF-PREDICTION COST

To validate whether self-prediction can estimate token cost at significantly lower expense, we compute the average over 4 independent OpenHands agent runs and 4 self-prediction runs, both using Claude Sonnet 3.7. Each configuration is evaluated on 500 samples independently, followed by statistical analysis. The results demonstrate that self-prediction achieves cost estimation with substantially lower mean total tokens (approximately $\times 4.5$ reduction) and lower mean cost (\$1.3814 vs \$6.1093). These findings confirm that self-prediction is a cost-efficient proxy for estimating token consumption.

Table 4: Comparison of Token Cost Statistics Between Ground Truth and Self-Prediction

| Statistic | Prompt Tokens | Completion Tokens | Total Tokens | Cost (\$) |
|---|---|---|---|---|
| **Ground Truth (Agent Call)** | | | | |
| Mean | 1,954,342 | 16,420 | 1,970,762 | 6.1093 |
| Std Dev | 2,064,830 | 12,931 | 2,077,229 | 6.3807 |
| Minimum | 221,670 | 3,472 | 225,141 | 0.7171 |
| 25th Percentile | 776,648 | 8,888 | 785,450 | 2.4622 |
| Median | 1,234,188 | 12,328 | 1,245,664 | 3.8950 |
| 75th Percentile | 2,374,227 | 19,377 | 2,390,590 | 7.3665 |
| Maximum | 16,304,326 | 116,109 | 16,420,436 | 50.6546 |
| **Prediction (Self-Prediction Call)** | | | | |
| Mean | 438,252 | 4,440 | 442,692 | 1.3814 |
| Std Dev | 125,800 | 1,076 | 126,270 | 0.3847 |
| Minimum | 149,042 | 1,761 | 153,291 | 0.4943 |
| 25th Percentile | 358,758 | 3,692 | 362,543 | 1.1405 |
| Median | 433,963 | 4,257 | 439,306 | 1.3746 |
| 75th Percentile | 510,467 | 4,974 | 515,511 | 1.6031 |
| Maximum | 1,066,478 | 9,951 | 1,075,758 | 3.3386 |

Meanwhile, we want to understand whether there is a correlation between the self-prediction costs and the actual costs. Figure 24 illustrates the regression fitting results. The statistical tests show a weak-to-moderate monotonic relationship, evidenced by:

- Pearson correlation: $r_p = 0.165137$

- Spearman correlation: $r_s = 0.353866$

Although the linear correlation remains weak ($r_p < 0.2$), the higher rank-based coefficient ($r_s > 0.35$) suggests that self-prediction is more effective at capturing relative ordering than precise magnitude estimation. This indicates that low-cost self-prediction can serve as a reasonable proxy to approximate *coarse-grained* trends in token cost, especially when exact token-level fidelity is not strictly required.

Further, we investigate whether the self-prediction cost impacts prediction accuracy. Using the same data conducted on Claude Sonnet 3.7, we compute the mean squared error (MSE) between predicted and ground-truth token counts for prompt tokens, completion tokens, and total tokens, respectively. We then quantify the association between self-prediction cost and prediction error using Pearson correlation analysis. As shown in the figure 25, the statistical results indicate that the correlation between self-prediction cost and prediction error is negligible (e.g., for total tokens, the Pearson correlation coefficient $r \approx 0.02$). This indicates that self-prediction cost and prediction accuracy are effectively independent, suggesting that token-cost self-prediction does not introduce bias into the accuracy of token consumption estimates and that the two factors can be regarded as mutually orthogonal.

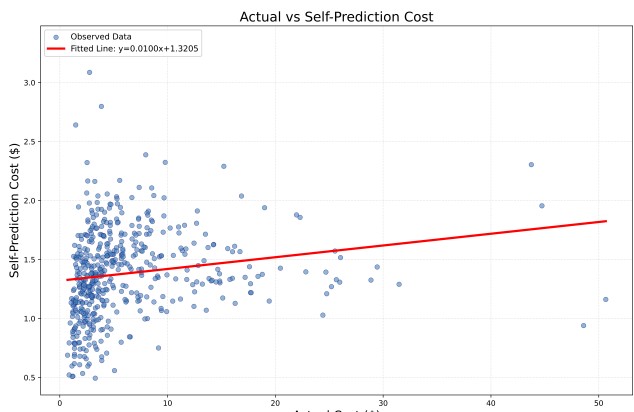

Figure 24: Correlation analysis between self-prediction and ground truth cost.

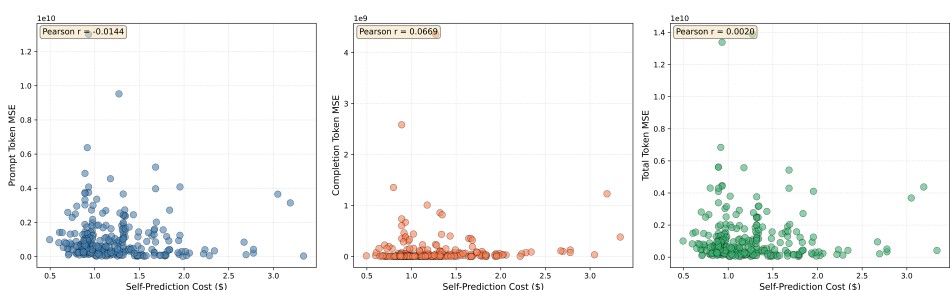

Figure 25: Correlation anslysis between self-prediction cost and prediction accuracy.

# D PROMPT FOR SINGLE LLM PREDICTION

## D.1 REPO SUMMARY

---

**Role**: REPOSITORYANALYZER — expert at analyzing code repositories and their structure.

**Task**: Analyze the provided repository file tree and problem statement, then create a concise summary capturing the key aspects relevant to solving the problem.

**Focus on:**

- **Repository Structure**: Key directories, file organization patterns
- **Technology Stack**: Programming languages, frameworks, build tools
- **Complexity Indicators**: Size, depth, and organization complexity
- **Problem-Relevant Files**: Files or directories most relevant to the task
- **Architecture Patterns**: Any clear architectural patterns or design principles

**Deliverable**: A structured, concise, and informative summary that helps someone understand the repository context for this specific problem.

---

**User Prompt Template (Summary Generation)**
*This is the concrete user-facing prompt assembled by your pipeline.*

```
Please analyze the following repository and problem statement to create
    a concise summary.

{<REPOSITORY_INFO_FORMATTED>}

Task: {<PROBLEM_STATEMENT>}
Difficulty Level: {<DIFFICULTY>}

Based on this information, provide a structured summary that captures
    the key aspects of the repository relevant to solving this problem.
    Focus on structure, technology stack, complexity, and problem-
    relevant components.
```

---

## D.2 PREDICTION AT DIFFERENT INFORMATION LEVEL

**Setting**: ZERO_PS_TOOL_DIFF_REPO

**Role**: TOKENESTIMATOR (not a coding agent).

**Always do exactly four things, in order:**

1. **Repository Context Analysis**: Review the provided repository summary and understand the codebase structure and complexity.

2. **Task Planning**: Based on the repository context and the specific task, outline a high-level approach for solving the problem.

3. **Token and Tool Estimation**: Estimate how many tokens the assistant *would* emit and how many tool calls it *would* invoke to solve the task, taking into account the difficulty level and repository structure. For each step, provide detailed reasoning for your estimates. Additionally, for tool use, break down how many times *each specific tool* would likely be invoked (e.g., `execute_bash`: 10, `str_replace_editor`: 20, `think`: 5).

4. **Output the following five lines in this exact format (and nothing else):**

```
<INPUT_TOKEN_ESTIMATE>####</INPUT_TOKEN_ESTIMATE>
<OUTPUT_TOKEN_ESTIMATE>####</OUTPUT_TOKEN_ESTIMATE>
<TOOL_USE_EXECUTE_BASH_ESTIMATE>##</TOOL_USE_EXECUTE_BASH_ESTIMATE>
<TOOL_USE_STR_REPLACE_EDITOR_ESTIMATE>##</
    TOOL_USE_STR_REPLACE_EDITOR_ESTIMATE>
<TOOL_USE_THINK_ESTIMATE>##</TOOL_USE_THINK_ESTIMATE>
```

**System Context Provided:**

- OpenHands capabilities and workflow:

  `{OPENHANDS_SYSTEM_PROMPT}`

- Available tools:

  `{TOOL_USE_PROMPT}`

**Restrictions**: Do not execute the task, do not call any tool, do not write code.

**Setting**: FEW_PS_TOOL_DIFF_REPO

**Role**: TOKENESTIMATOR (not a coding agent).

**Always do exactly four things, in order:**

1. **Repository Context Analysis**: Review the provided repository summary and understand the codebase structure and complexity.

2. **Task Planning**: Based on the repository context and the specific task, outline a high-level approach for solving the problem.

3. **Token and Tool Estimation**: Estimate how many tokens the assistant *would* emit and how many tool calls it *would* invoke to solve the task, taking into account the difficulty level and repository structure. **Use the provided examples to inform your estimates** and provide detailed reasoning for each step. Additionally, for tool use, break down how many times *each specific tool* would likely be invoked (e.g., `execute_bash`: 10, `str_replace_editor`: 20, `think`: 5).

4. **Output the following five lines in this exact format (and nothing else):**

```
<INPUT_TOKEN_ESTIMATE>####</INPUT_TOKEN_ESTIMATE>
<OUTPUT_TOKEN_ESTIMATE>####</OUTPUT_TOKEN_ESTIMATE>
<TOOL_USE_EXECUTE_BASH_ESTIMATE>##</TOOL_USE_EXECUTE_BASH_ESTIMATE>
<TOOL_USE_STR_REPLACE_EDITOR_ESTIMATE>##</
    TOOL_USE_STR_REPLACE_EDITOR_ESTIMATE>
<TOOL_USE_THINK_ESTIMATE>##</TOOL_USE_THINK_ESTIMATE>
```

**Examples Section (when few-shot examples are available):**

```
Below are examples that can help you understand the task and better
    estimate the token usage:

Example i:

Repository Summary:
{ex["repo_summary"]}

Task: {ex["problem_statement"]}
Difficulty Level: {ex["difficulty"]}

<INPUT_TOKEN_ESTIMATE>####{ex["total_prompt_tokens_mean"]}</
    INPUT_TOKEN_ESTIMATE>
<OUTPUT_TOKEN_ESTIMATE>####{ex["total_completion_tokens_mean"]}</
    OUTPUT_TOKEN_ESTIMATE>
<TOOL_USE_EXECUTE_BASH_ESTIMATE>##{ex["avg_tool_usage_execute_bash"]}</
    TOOL_USE_EXECUTE_BASH_ESTIMATE>
<TOOL_USE_STR_REPLACE_EDITOR_ESTIMATE>##{ex["
    avg_tool_usage_str_replace_editor"]}</
    TOOL_USE_STR_REPLACE_EDITOR_ESTIMATE>
<TOOL_USE_THINK_ESTIMATE>##{ex["avg_tool_usage_think"]}</
    TOOL_USE_THINK_ESTIMATE>
```

**System Context Provided:**

- OpenHands capabilities and workflow:

  `{OPENHANDS_SYSTEM_PROMPT}`

- Available tools:

  `{TOOL_USE_PROMPT}`

**Restrictions**: Do not execute the task, do not call any tool, do not write code.

# E    PROMPT FOR SELF-PREDICTION BY THE SAME AGENT

## E.1    ZERO-SHOT SETTING

**IMPORTANT**: You are a **TOKEN ESTIMATION** agent, NOT a problem-solving agent. Your ONLY goal is to estimate token costs, NOT to fix bugs or implement features. You MUST call the `finish` tool with a JSON estimate, NEVER with actual code changes.

Your task is to estimate how many LLM tokens would be consumed to solve this problem if a coding agent were to complete it end-to-end.

**Follow these phases to estimate token costs:**

**Phase 1. EXPLORATION**: Explore the codebase to understand the problem

- 1.1 Read the problem description and understand what needs to be fixed
- 1.2 Explore relevant files and directories to understand the codebase structure
- 1.3 Search for key functions, classes, or variables related to the issue
- 1.4 Identify the root cause and complexity of the problem

**Phase 2. ANALYSIS**: Analyze the complexity and required changes

- 2.1 Assess the scope of changes needed (number of files, lines of code)
- 2.2 Consider the debugging iterations likely needed
- 2.3 Evaluate the testing complexity and iterations
- 2.4 Estimate the number of tool calls and reasoning steps

**Phase 3. TOKEN ESTIMATION**: Calculate token usage for the complete solution

- 3.1 Estimate input tokens for:
    - Repository exploration and file reading
    - Code analysis and debugging
    - Implementation iterations
    - Testing and verification
- 3.2 Estimate output tokens for:
    - Reasoning and analysis responses
    - Code generation and explanations
    - Debugging responses
    - Test results interpretation
- 3.3 Calculate total tokens and confidence level

**Phase 4. FINISH**: Provide final token estimate

- 4.1 Call the `finish` tool with a JSON object containing:
    - `predicted_input_tokens`
    - `predicted_output_tokens`
    - `predicted_total_tokens`
    - `confidence` (0--1)
    - `breakdown_by_phase`

**Remember:** You are estimating **COSTS**, not implementing SOLUTIONS. Do **NOT** write actual code fixes or modify any files. Your final deliverable is a **JSON token estimate**, not a working solution.

## E.2 ZERO-SHOT + FINE-GRAINED BREAKDOWN SETTING

> **IMPORTANT**: You are a **TOKEN ESTIMATION** agent, NOT a problem-solving agent. Your ONLY goal is to estimate token costs, NOT to fix bugs or implement features. You MUST call the `finish` tool with a JSON estimate, NEVER with actual code changes.
>
> **Fine-Grained Requirements:**
>
> - Round all estimates to the nearest **10 tokens** (e.g., 30, 40, 70). Do NOT round to hundreds.
> - Break down into **atomic actions**: reading issue description, listing files, inspecting one file, analyzing one function/class, one debugging iteration, one test run.
> - Do NOT output ranges (e.g., "200–300"). Always a single number.
> - First give sub-action estimates, then sum into totals per phase, then overall totals.
> - Final JSON must contain fine-grained estimates inside `breakdown_by_phase`.
>
> (Other parts are the same as in the zero-shot setting and are omitted.)

## F   TOKEN CONSUMPTION FOR SELF-PREDICTION BY THE SAME AGENT

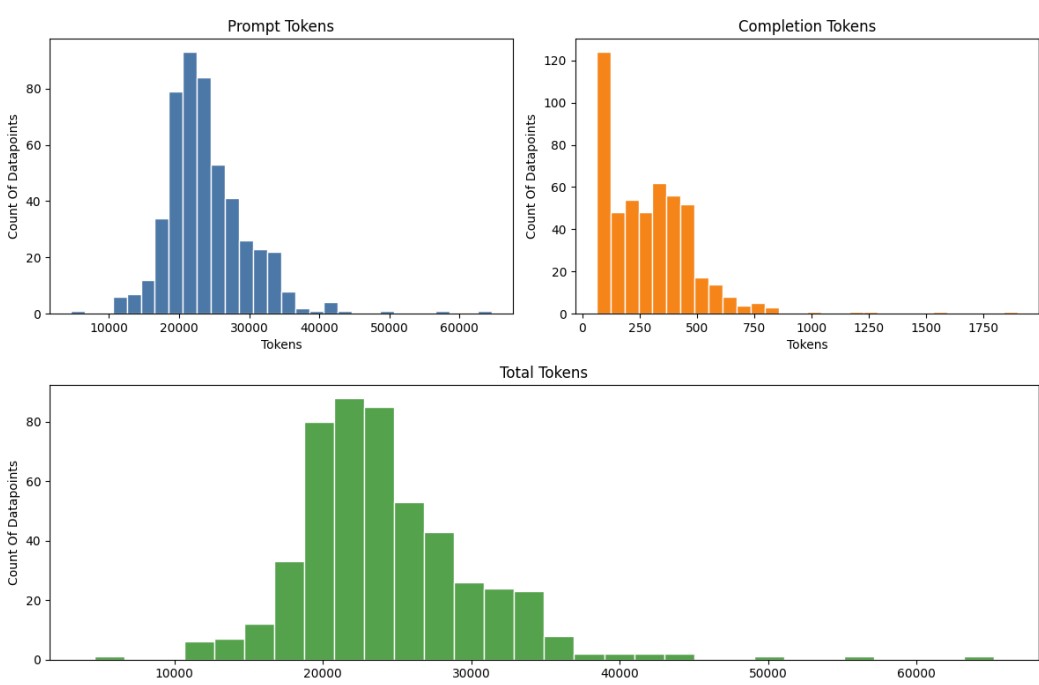

Figure 26: **Token consumption for self-prediction within one run.** This figure shows the prompt, output, and total tokens consumed during a single 500-sample self-prediction run. The estimation process alone consumes a significant number of tokens, which is an important overhead to consider.

## G   THE USE OF LARGE LANGUAGE MODELS (LLMS)

We used Claude Sonnet 3.7 as the main model to predict token consumption in our experiments. In addition, large language models (e.g., ChatGPT) were used only as general writing aids to polish language, improve clarity, and format prompts. All experimental design, data analysis, and conclusions were made and verified by the authors.

