# OpenReview forum: "How Do Coding Agents Spend Your Money? Analyzing and Predicting Token Consumptions in Agentic Coding Tasks"
_ICLR.cc/2026/Conference — ICLR 2026 Conference Withdrawn Submission_

### Official Review · Reviewer_mggr · 2025-10-25

**Soundness:** 2
**Presentation:** 2
**Contribution:** 2
**Rating:** 2
**Confidence:** 3

**Summary:**

This paper addresses a timely and underexplored aspect of AI agents: the analysis and prediction of token consumption in agentic coding workflows. Using trajectories from Claude Sonnet 3.7 on the SWE-bench dataset, the authors provide empirical insights into token usage patterns and evaluate prediction methods. While the work offers novel findings on cost dynamics and highlights practical challenges, it is limited by its reliance on a single model and proprietary pricing schemes, which undermines generalizability. The prediction results are underwhelming, raising questions about the feasibility of the proposed approaches. Overall, this is an exploratory study with interesting observations, but it requires broader experimentation to strengthen its claims. I recommend rejection because it needs to address generalizability and overstatements of prediction accuracy.

**Strengths:**

The lack of transparency in AI agent pricing is a significant barrier to their widespread adoption. As the authors claim, this appears to be the first empirical study on agent token consumption. By focusing on agentic coding tasks (e.g., via SWE-bench), the paper potentially informs better pricing models and user expectations.

The core discovery that higher token costs are associated with lower accuracy (termed the "inverse test-time scaling paradox"), is compelling and counterintuitive. This challenges assumptions about scaling test-time compute and suggests inefficiencies in agent trajectories, such as excessive exploration leading to diminished performance.

The authors meticulously decompose token types (e.g., non-cached prompts, cache creation, cache reads, and completion tokens) and their contributions to costs across different phases (early, mid, and late).

**Weaknesses:**

The entire empirical analysis and prediction experiments are based solely on a single, closed-source LLM Claude Sonnet 3.7. Although the authors acknowledge this in the limitations section, it severely restricts the paper's scope. The title implies broad applicability to "AI Agents," but the conclusions may not extend to other models (for example, GPT-series or open-source alternatives). A more accurate framing might be "How Does Claude Sonnet 3.7 Spend Your Money?" to reflect the narrow focus.

The cost analysis (particularly in Figures 3 and 4) heavily depends on Anthropic's specific pricing scheme, which differentiates rates for cache creation, cache reads, and non-cached inputs. Switching to a different provider (e.g., GPT-4o) or an open-source model could yield entirely different cost dynamics. The methodology fails to decouple these findings from vendor-specific decisions, limiting the transferability of the results.

The latter half of the paper focuses on token prediction, but the core results indicate failure: Pearson's correlation for total token consumption is below 0.15, which is essentially negligible. The authors claim that predicting output tokens and ranges is "practical and reasonably accurate," but Figure 5 shows that even the best setting (PTDR with LogScale-FewShot) achieves only around 0.36 correlation for output tokens, a value typically considered "weak" to "moderate" in prediction tasks. Describing this as "reasonably accurate" overstates the findings and could mislead readers.

The authors do not evaluate the real-world utility of "self-prediction" approaches, such as the overhead costs involved. If a task requires only 50,000 tokens but the prediction process consumes 30,000 tokens, users end up paying for 80,000 tokens total, rendering the method economically unviable. Additionally, the noted tendency for self-prediction to "overestimate true costs" further diminishes its practicality, yet this is not critically discussed.

**Questions:**

The "inverse test-time scaling" finding is intriguing. Could this be linked to specific agent behaviors, like looping in exploration? Are there ablation studies on agent prompts or tools that might mitigate this?

---

> ### Author Response · Authors · 2025-12-02
>
> Thank you for your thoughtful and constructive feedback. We appreciate your recognition of the novelty of our study and all the valuable suggestions! Please see our responses to your questions below:
>
> W1: Reliance on a single model (Claude Sonnet 3.7)
>
> Thanks for pointing this out. To assess generalizability across models, we additionally ran experiments with Claude Sonnet 4, Qwen3-Coder-480B/A35B-Instruct, and GPT-5, and we include these results in the revised appendix (section A in appendix, line 648-983). These models span different families (different companies, open-source and closed-source), use different caching mechanisms, and follow different pricing schemes, yet all exhibit the same qualitative patterns across our analyses.
>
> W2: Dependence on a single charging scheme
>
> Thanks a lot for raising this concern. As discussed in W1, we have included additional analysis on other models and the results remain generally consistent. While different models have different charging schemes, the types of token charges are generally consistent (e.g. input tokens, output token, caching tokens) and different types of token consumptions are the key factor that’s driving the final costs instead of the specific charging scheme. From the additional experiment results, we can observe different charging patterns, but the core behavioral findings remain stable, for example, the input/cache token dominating, inverse test-time scaling paradox generally holds across all models.
>
> W3: Concern about overstating prediction performance
>
> We appreciate this feedback. Our intention is not to claim a fully solved prediction method. Instead, our contribution is to identify, formalize, and empirically characterize the underlying phenomena: input-token dominance, inverse test-time scaling, and the inherent difficulty of predicting total token consumption. This positions the paper as an analysis and problem-definition study, similar to early descriptive studies that preceded research on scaling laws or retrieval behavior, etc.
>
> We are sorry for the confusion caused by our initial framing of the results and we have revised the sections accordingly. See line 024-028
>
> W4: discussion over self-prediction method
>
> We agree that it is important to analyze the relationship between ‘prediction cost’ and actual ‘task-solving cost’.
>
> We choose the self-prediction method because accurate cost prediction requires access to a wide range of internal signals: agent architecture, backbone model behavior, task statement, repository structure, tool use patterns, and more. The most promising signal came from allowing the agent to reason dynamically about the task and repository itself, an approach that aligns better with how such systems are expected to operate in practice.
>
> To conduct such an analysis, we computed both the correlation between prediction cost and task-solving cost and also between prediction cost and prediction accuracy, and discussed the results in the paper (Section C in appendix, Line 1080-1160, Line 450-459). The results showed that the cost for self-prediction is weakly correlated with the task actual cost, but substantially lower. Meanwhile, the correlation between self-prediction cost and prediction error is negligible.

---

> > ### Author Response · Authors · 2025-12-02
> >
> > Q1: more explanation over "inverse test-time scaling"
> >
> > Thanks for pointing out this insightful question. This is indeed a key mechanism underlying agent failures which is important towards understanding and designing better agents.
> >
> > Qualitatively, our inspection of trajectories shows that high-cost failures often involve repetitive behaviors such as repeatedly opening the same file or performing multiple ineffective edits, which greatly increase context length and degrade performance.
> >
> > Quantitatively, we quantify latent behaviors like repeated file editing/checking to make the observation more solid. We included such discussion in Section 3.1 Line 193-205, we observed that many expensive but failed runs are characterized by redundant back-and-forth file access and re-editing. When there are less increases in repeated file operations as cost rises, the ‘inverse test-time scaling’ will also be weaker.
> >
> >
> > Mitigating such behaviors is an important direction for follow-up work, but as discussed in W3, this paper focuses more on surfacing the underlying phenomenon.

---

### Official Review · Reviewer_mTgG · 2025-10-31

**Soundness:** 3
**Presentation:** 3
**Contribution:** 2
**Rating:** 4
**Confidence:** 4

**Summary:**

This paper presents the first empirical study to analyze and predict the token consumption of AI agents in complex coding tasks. The authors identify that unpredictable and often massive token usage is a major barrier to the adoption of agentic systems, creating non-transparent pricing and risk for users .

The paper's contributions are:


An empirical analysis of agent trajectories from the SWE-bench dataset. This analysis reveals several key findings: (a) Token consumption has extremely high variance, with some runs using 10x more tokens than others for the same task. (b) More token usage is correlated with lower accuracy, a phenomenon the authors call an "inverse test-time scaling paradox" . (c) Unlike chat models, input tokens dominate overall consumption and cost in coding agents, even with caching.





A prediction task formulation to estimate token consumption before task execution. The authors benchmark multiple prediction methods, including zero-shot and few-shot prompting of a single LLM, as well as letting the agent "self-predict" its own costs.


The paper concludes that while precisely predicting the total token consumption is extremely challenging (Pearson's r < 0.15), predicting output token counts and the log-scale range of total consumption is feasible and reasonably accurate.

**Strengths:**

The paper tackles a practical, high-impact problem (cost prediction) that is a major barrier to real-world agent adoption.

The "inverse scaling" paradox and the "input token dominance" are both major, counter-intuitive findings that will directly influence future research into agent efficiency.

 The analysis is based on strong, relevant data (SWE-bench) and the methodology is rigorous.

The paper provides immediate, actionable advice for agent providers (e.g., "offer log-scale ranges or budget alerts, not exact quotes") and for researchers (e.g., "focus on optimizing input token ingestion, not just generation").

**Weaknesses:**

1. The entire paper's analysis is based on one model (Claude Sonnet 3.7), one framework (OpenHands), and one task domain (coding on SWE-bench) . This is a significant limitation.



2. Do the findings generalize? Does the "inverse scaling" paradox hold for GPT-4o or Llama 3.1? Does "input token dominance" hold for other input-heavy tasks, like document analysis or web research, or is it unique to coding? The paper claims to analyze "agentic coding tasks", but it has only analyzed one agent's behavior.

3, While a negative result is a valid scientific contribution, the fact remains that the primary goal of predicting total token cost was not achieved (r < 0.15). This makes the paper more of an "analysis and problem-definition" paper than a "solution" paper.


4. Cost of Self-Prediction: The paper rightly explores "self-prediction" but Figure 7 shows this prediction process is itself very expensive, consuming a median of ~25,000 prompt tokens and ~400 completion tokens. This high cost to get a prediction makes the method impractical. The paper notes this as a challenge but doesn't fully analyze this trade-off (e.g., is the prediction cost correlated with the actual cost?).

**Questions:**

1. The paper's most significant findings (inverse scaling, input-token dominance) are based on a single model and benchmark. How confident are you that these findings are fundamental to agentic coding, rather than being an artifact of the OpenHands framework or Claude 3.7's specific reasoning patterns?

2. The "Inverse Scaling" Paradox: This is a fascinating finding. What is your hypothesis for the cause of this? The paper suggests "inefficient trajectories", but could it also be that longer context histories (from more tokens) actively degrade the agent's reasoning, causing it to get "lost in the middle" and fail? I am interested in it.

3. The "self-prediction" agent itself consumes a significant number of tokens to produce an estimate (shown in Figure 7). Did you find any correlation between the cost of the prediction and the actual cost of the task? It seems this prediction overhead makes the method impractical for all but the most expensive tasks.

4. Why was predicting input tokens so much harder than output tokens (Figure 5)? Is it because the agent's exploration path (which files to read, which tools to call) is fundamentally more stochastic than its generation (which just solves a given sub-problem)?

5. Your paper's key finding is that input tokens—from repository exploration and tool use—are the primary cost driver, not completion tokens. one paper AgentTaxo (https://openreview.net/pdf?id=0iLbiYYIpC) independently discovers this exact same phenomenon in multi-agent systems, finding that input tokens outnumber output tokens by 2:1 to 3:1 due to inter-agent communication. Given that both papers are pioneering the empirical analysis of agent tokenomics and have validated this same counter-intuitive finding in complementary domains (agent-to-environment vs. agent-to-agent), can you please address this gap by citing AgentTaxo and discussing how your work and theirs together provide a more complete picture of why agentic systems are so input-heavy?

---

> ### Author Response · Authors · 2025-12-02
>
> Thank you for your thoughtful and constructive feedback. We appreciate your recognition of the novelty and practical impact of our study, as well as your detailed questions that help us clarify and strengthen the paper.
>
> W1: One model, one framework, one benchmark
>
> Thank you for raising this important point.
>
> For the dataset we choose SWE-bench-verified as this is the only large-scale software engineering dataset that has been rigorously validated by human annotators to exclude problematic or ambiguous cases, making it uniquely suitable for careful empirical analysis.
>
> We choose OpenHands as the framework also because it provides state-of-the-art performance among open-weight agent frameworks and offers transparent scaffolding and fully auditable execution, which is crucial for a large-scale study focused on token-level behavior.
>
> To assess generalizability across models, we additionally ran experiments with Claude Sonnet 4, Qwen3-Coder-480B/A35B-Instruct, and GPT-5, and we include these results in the revised appendix A (Line 648-983). These models span different families (different companies, open-source and closed-source), use different caching mechanisms, and follow different pricing schemes, yet all exhibit the same qualitative patterns across our analyses.
>
> We acknowledge that broader benchmarks and frameworks would be ideal, but collecting multi-round agent trajectories is extremely time- and resource-intensive, which limits the feasible scope for this first empirical study.
>
>
> W2: The paper claims to analyze "agentic coding tasks", but it has only analyzed one agent's behavior.
>
> We agree this should be positioned more clearly as a case study, serving as the first empirical examination of token consumption in agentic coding. We have clarified our claims in the revised version (Line17-18). Mainwhile, as agent trajectories typically include dozens of interaction rounds, our statistics show that running a single agent inference and evaluation on SWE-bench Verified can cost several hundred dollars, making additional agents and benchmarks economically unaffordable for us in practice.
>
> W3: While a negative result is a valid scientific contribution, the fact remains that the primary goal of predicting total token cost was not achieved (r < 0.15). This makes the paper more of an "analysis and problem-definition" paper than a "solution" paper.
>
> Thanks a lot for pointing this out! Our goal is not to present a fully solved prediction method, but rather to identify, formalize, and empirically characterize the phenomena underlying token consumption. Similar to early descriptive studies that preceded research on scaling laws or retrieval behavior, our contribution lies in surfacing fundamental patterns: heavy-tailed usage, input-token dominance, inverse test-time scaling, and prediction limits, which can motivate future algorithmic work.
>
>
> W4: prediction cost correlated with the actual cost
>
> Thanks for pointing out this important question. We acknowledge the importance of such analysis. To conduct such an analysis, we computed both the correlation between prediction cost and task-solving cost and also between prediction cost and prediction accuracy, and discussed the results in the paper (Section C in appendix, Line 1080-1160, Line 450-459). The results showed that the cost for self-prediction is weakly correlated with the task actual cost, but substantially lower. Meanwhile, the correlation between self-prediction cost and prediction error is negligible.

---

> > ### Author Response · Authors · 2025-12-02
> >
> > Q1: Single model and benchmark, the generalization of the results.
> >
> > We have experimented with additional models and observe similar trends for all the key findings. Please see the discussion in W1 for more details.
> >
> > Q2: reason for "Inverse Scaling" Paradox
> >
> > Thanks for pointing out this insightful question. This is indeed important towards understanding and designing better agents.
> >
> > Qualitatively, our inspection of trajectories shows that high-cost failures often involve repetitive behaviors such as repeatedly opening the same file or performing multiple ineffective edits, which greatly increase context length and degrade performance.
> >
> > Quantitatively, we quantify latent behaviors like repeated file editing/checking to make the observation more solid. We included such discussion in Section 3.1 Line 193-205, we observed that many expensive but failed runs are characterized by redundant back-and-forth file access and re-editing. When there are less increases in repeated file operations as cost rises, the ‘inverse test-time scaling’ will also be weaker.
> >
> > Q3: Correlation between the cost of the prediction and the actual cost of the task
> >
> > Please see the discussion in W4.
> >
> > Q4: Why was predicting input tokens so much harder
> >
> > There are two key reasons:
> >
> > 1. Framework design: The agent appends its entire conversation history including all tool responses into each new prompt. This makes input length grow almost linearly and is difficult to predict at once, whereas output length remains more stable.
> >
> > 2. File ingestion: When the agent reads files or diffs, the entire content is injected into the prompt. These context expansions depend on the agent’s exploration path and are difficult to infer from the static problem description.
> >
> > This intrinsic variability makes input-token prediction significantly more challenging.
> >
> >
> > Q5: cite the paper and add analysis
> >
> > Thank you for the suggestion. We agree this strengthens the framing. We have incorporated a citation and added discussion in the introduction (Line 58–60). Together, the two studies provide a broader and more unified picture of why agentic systems are heavily input-driven.

---

### Official Review · Reviewer_2jw3 · 2025-10-31

**Soundness:** 3
**Presentation:** 3
**Contribution:** 2
**Rating:** 4
**Confidence:** 3

**Summary:**

This paper systematically investigates the token consumption patterns and cost predictability of coding agents driven by large language models (LLMs) when performing real-world software engineering tasks on the SWE-bench Verified dataset.

**Strengths:**

This is one of the first studies focusing on token-level cost analysis for LLMs. I find some of its conclusions particularly insightful for users who are attentive to token cost, especially when optimizing prompt design or inference budgeting.

**Weaknesses:**

1. Since this is an empirical analysis paper, it is important to justify the sufficiency and generalizability of using SWE-bench as the sole benchmark across all dimensions of investigation. To what extent do these conclusions generalize to other platforms?
2. The “accuracy-cost trade-off frontier” is not explored.
3. The observed negative correlation between token usage and accuracy may stem from problem difficulty, as the paper does not control for this variable.

**Questions:**

1. It would be better to expand the experimental datasets to include more diverse programming and reasoning tasks.
2. I am wondering if this token-consumption problem can be solved by optimization techniques.
3. I would suggest to establish a performance-cost trade-off curve to better quantify efficiency and accuracy relationships.

---

> ### Author Response · Authors · 2025-12-02
>
> Thank you for your encouraging feedback and the recognition of the novelty of our study, and the insights we conveyed.
>
>
> W1: Justification of the sufficiency and generalizability of using SWE-bench as the sole benchmark
>
> Thank you for raising this important point.
>
> To the best of our knowledge, among publicly available, widely-used benchmarks, SWE-bench-Verified remains the most comprehensive that combines repository-level context, executable validated patches and real issue distributions. Although recent benchmarks such as FEA-Bench attempt to broaden coverage and mitigate limitations, they have not yet reached the combined level of scale, task diversity, and widespread adoption demonstrated by SWE-bench.
>
> We choose OpenHands as the framework also because it provides state-of-the-art performance among open-weight agent frameworks and offers transparent scaffolding and fully auditable execution, which is essential for our large-scale empirical study.
>
> To assess generalizability across models, we additionally ran experiments with Claude Sonnet 4, Qwen3-Coder-480B/A35B-Instruct, and GPT-5, and we include these results in the revised appendix (Section A in appendix, Line 648-983). These models span different families (different companies, open-source and closed-source), use different caching mechanisms, and follow different pricing schemes, yet all exhibit the same qualitative patterns across our analyses.
>
> We acknowledge that broader benchmarks and frameworks would be ideal, but collecting multi-round agent trajectories is extremely time- and resource-intensive (one single run will cost hundreds of dollars), which limits the feasible scope for this first empirical study.
>
>
> W2: Accuracy/performance–cost trade-off; request for trade-off curves
>
> Thank you for pointing out this important aspect. In the revised version, we now explicitly discuss the accuracy–cost trade-off. As shown in Fig. 2a and discussed in Line165–205, accuracy decreases as token usage increases across tasks. Further, Fig. 2b shows accuracy–cost trends within the same problem: runs grouped from MinCost to MaxCost consistently exhibit an “inverse test-time scaling” pattern, with accuracy peaking at moderate cost and dropping sharply for the highest-cost runs. Appendix results confirm that this trend holds across multiple backbone models.
>
>
> W3: Negative correlation between token usage and accuracy may stem from problem difficulty.
>
> Thank you for pointing this out! We totally agree that this might be the case. However, in our experiments, we not only studied the cross-task setting, but also studied runs for the same task. Fig 2b (Line 177-191) shows the results for cost trends within each problem and we observe that  even under identical problem difficulty, accuracy rises slightly at lower token cost but falls significantly when the token cost is high, demonstrating that difficulty alone cannot explain the negative accuracy–cost relationship.

---

> > ### Author Response · Authors · 2025-12-02
> >
> > Q1: Expand datasets to include diverse programming and reasoning tasks
> >
> > Thanks a lot for this suggestion. We have discussed our motivation behind using SWE-bench-verified in the response to W1. In our initial exploration, we experimented with various reasoning tasks. However, the token consumption for many pure reasoning tasks are very limited and therefore relatively easy to predict. We chose agentic coding tasks as it involves complex planning and executions and as it is one of the most widely used agentic tasks in the real world.
> >
> >
> > Q2: Whether token consumption can be optimized through algorithmic or learned techniques
> >
> > Thank you for your thoughtful question. We did explore learning-based approaches using an external model with optimization techniques, but ultimately decided not to focus on them in the main paper for the following reasons:
> >
> > First, agentic coding is closely tied to real deployment scenarios, where relying on an external model to estimate token cost is often impractical. In real-world use, the agent is expected to anticipate its own cost before execution, much like a software engineer estimating their own time. This is why we focus on prompting the same backbone model or letting the agent perform the prediction itself, which better mirrors real-world agent workflows.
> >
> > Second, accurate cost prediction requires access to a wide range of internal signals: agent architecture, backbone model behavior, task statement, repository structure, tool use patterns, and more. We attempted to use Gemini embedding and a RoBERTa-based model to encode static features at various levels of richness and trained regression heads for prediction. However, performance was close to random. The most promising signal came from allowing the agent to reason dynamically about the task and repository itself, an approach that aligns better with how such systems are expected to operate in practice.
> >
> >
> > Q3: Establishing a performance–cost trade-off curve
> >
> > Please see our responses in W2 and W3. We now include explicit figures (fig2 and fig3) and discussion analyzing performance–cost trade-offs both across tasks and within tasks (line 165-205). We observed that the performance does not always increase as cost increases, instead, in many cases, the increased cost would be associated with redundant behaviors and hurt the performance.

---

### Official Review · Reviewer_tDsj · 2025-11-08

**Soundness:** 2
**Presentation:** 3
**Contribution:** 2
**Rating:** 2
**Confidence:** 3

**Summary:**

This paper makes use of LLM trajectories on SWE-bench to study how coding agents consume tokens for different tasks and explore the possibilities of predicting the execution cost. The problem is interesting and important given the recent proliferation of agentic AI. Also, some of the findings based on the token statistics of the agents are interesting. Yet all the prediction tasks and analytics tasks are carried out using prompts, and no learning algorithms are involved in this paper. Seems that its current contents better fit some other conferences than this conference focusing on representation learning.

**Strengths:**

The problem based on the SWE-bench data is interesting and could trigger more learning tasks related to Agentic AI to be formulated. The paper is asking some interesting questions.

**Weaknesses:**

The technical contribution of this paper is on making use of the trajectory data and carrying out some analytics tasks to answer the questions raised related to token consumption by coding agents. It does not propose any algorithm or formulation which are related to representation learning tasks. Even though I like the questions being asked, the discussion with details related to agentic AI is only the illustration of the prediction task instead of some research roadmap.

**Questions:**

Q1: Other than using LLMs to predict the execution cost, why not trying to learn some prediction models?

Q2: Given the findings of the paper, can you see some learning tasks which can be defined? E.g., given that the same task could result in big differences in token consumption, other than just warning the user, can the task execution and/or input token for context ingestion be analyzed and optimized first (like query optimization in DBMS)? This is only one example. Will there be more related learng tasks related to the usage data which can be identified and formulated?

---

> ### Author Response · Authors · 2025-12-02
>
> Thank you for your thoughtful feedback on our paper. We appreciate your recognition of the importance of studying token consumption in agentic coding systems and your interest in the questions we raise. Below we address your concerns in detail.
>
>
> W1: The paper does not propose any algorithm or formulation which are related to representation learning tasks. Even though I like the questions being asked, the discussion with details related to agentic AI is only the illustration of the prediction task instead of some research roadmap.
>
> Thank you for pointing this out. Our goal in this work is to establish the first systematic empirical foundation for understanding token consumption in agentic coding. The purpose of this paper is to identify, formalize, and empirically characterize the phenomena to inform the design of representation-learning or optimization methods.
>
> Similar to early empirical analyses that preceded major research threads (e.g., scaling laws, retrieval-augmented generation behaviors), our intention is to surface fundamental patterns: high variance of each run, heavy-tailed usage, input-token dominance, inverse test-time scaling, and predictability limits, which can later motivate more algorithmic developments.
>
>
> Q1: Other than using LLMs to predict the execution cost, why not try to learn some prediction models?
>
> Thank you for your thoughtful question. We have experimented with several external models (for example: the Gemini generated embedding and RoBERTa based scoring model, and also MLP as regressior) for this task in our initial experiments, however, we didn’t include the results because of the following reasons:
>
> First, agentic coding is closely tied to real deployment scenarios, where relying on an external model to estimate token cost may not be realistic. In practice, the agent itself must anticipate its own cost before execution, just as a software engineer estimates their own time. This is why we focus on prompting the same backbone model or letting the agent itself for prediction, which better mimics the real-world agent workflow.
>
> Second, accurate prediction requires a large amount of internal information: details about the agent framework, backbone model behavior, problem statement, repository structure, tool-use patterns, and more. We attempted to use Gemini embedding and a RoBERTa-based model to encode different levels of static information and trained regression heads for prediction, but performance remained close to random. The most promising signal comes from allowing the agent itself to reason dynamically about the repository and task, which aligns with how real systems operate.
>
>
> Q2: More discussions than only prediction tasks, discuss other research maps, more related learning tasks related to usage data.
>
> We appreciate your insightful suggestion. Our study naturally opens up the opportunities for many learning tasks. For example, how to have the model learn to predict its own token costs, how to train the model so that they are more budget-aware. We have actually started to explore these tasks as new projects.
>
> However, the scope of our work is to provide a systematic empirical analysis to set up this new problem of token consumption prediction. While we do not explore agent-framework optimization or design new learning algorithms in this paper, we believe our analysis lays the groundwork for such directions.

---

### Author Response · Authors · 2025-12-03
**Summary of Revisions and Clarifications for the Area Chair**

We sincerely thank the Area Chair for their time, careful consideration, and service to the community in coordinating the review of our work. We also greatly appreciate the constructive feedback and the effort invested in evaluating the submission.

Our paper presents the first systematic empirical study of token consumption in agentic coding systems, focusing on real-world software engineering tasks from SWE-bench-Verified. Across reviewers, several common concerns emerged regarding scope, generalizability, prediction methodology, and interpretation of the key phenomena. Below we summarize these points and how the revised paper addresses them.

1. The position of our paper

Our goal of this paper is not to present a fully solved prediction method or improved agent framework, but rather to identify, formalize, and empirically characterize the phenomena underlying token consumption. Similar to early descriptive studies that preceded research on scaling laws or retrieval behavior, our contribution lies in surfacing fundamental patterns: heavy-tailed usage, input-token dominance, inverse test-time scaling, and prediction limits, which can motivate future algorithmic work.


2. The generalizability of our empirical results

We appreciate the reviewers raising the importance of generalizability.

Among publicly available benchmarks, SWE-bench-Verified remains the most comprehensive setting that combines repository-level context, executable validated patches, and real-world issue distributions.

OpenHands was chosen for its state-of-the-art open-weight performance and fully auditable agent scaffolding, which is essential for large-scale token-level analysis.

To assess model generalization, we additionally evaluated Claude Sonnet 4, Qwen3-Coder-480B/A35B-Instruct, and GPT-5 in the revised appendix. Despite spanning different providers, architectures, and pricing mechanisms, all reproduce the same qualitative patterns observed in the main paper.

While broader benchmarks and frameworks would be ideal, the high cost (hundreds of dollars for one run) of collecting multi-round agent trajectories limits the feasible scope of this first study.


3. Learning- or Optimization-Based Prediction Methods

We explored external learned predictors, including Gemini embeddings, RoBERTa-based scorers, and MLP regressors, but found their performance to be near random.

More importantly, in real deployments, cost estimation must be performed by the agent itself rather than by an external model with privileged access. Accurate prediction requires rich internal signals such as repository exploration, tool usage, and dynamically evolving context, which static encoders cannot capture. Even if one were to train an external model that perfectly fits a specific agent, model, and framework, such a predictor would still face severe generalization challenges when transferred to new architectures, tools, or environments. This motivates our focus on prompting-based prediction and self-prediction.


4. More Analysis of the “Inverse Test-Time Scaling” Effect

Through both qualitative inspection and quantitative analysis, we find that high-cost failures are dominated by inefficient behaviors such as repeated file openings and redundant edits, which inflate context length and degrade reasoning. Even within the same task, runs with excessive repeated file operations are significantly more likely to fail, showing that increased cost is not simply a proxy for problem difficulty. This provides a concrete behavioral explanation for the inverse test-time scaling phenomenon.


5. Analysis of Self-Prediction Overhead

We analyze the relationship between prediction cost and actual task-solving cost, as well as between prediction cost and prediction error. Self-prediction is motivated by the need for dynamic access to internal reasoning and exploration signals. Our results show that self-prediction cost is weakly correlated with actual task cost and substantially lower than full task execution, while being largely uncorrelated with prediction error.


Overall, we position this work as an empirical case study of agent token economics that establishes key phenomena, exposes intrinsic prediction limits, and sets the stage for future algorithmic optimization.

---

### Note · Authors · 2026-01-06

I have read and agree with the venue's withdrawal policy on behalf of myself and my co-authors.